# The role of interdecadal climate oscillations in driving Arctic atmospheric river trends

Weiming Ma [1] ✉, Hailong Wang [1] ✉, Gang Chen [2], L. Ruby Leung [1], Jian Lu [1], Philip J. Rasch [3], Qiang Fu [3], Ben Kravitz [1,4], Yufei Zou [1], John J. Cassano [5,6,7] & Wieslaw Maslowski [8]

Atmospheric rivers (ARs), intrusions of warm and moist air, can effectively drive weather extremes over the Arctic and trigger subsequent impact on sea ice and climate. What controls the observed multi-decadal Arctic AR trends remains unclear. Here, using multiple sources of observations and model experiments, we find that, contrary to the uniform positive trend in climate simulations, the observed Arctic AR frequency increases by twice as much over the Atlantic sector compared to the Pacific sector in 1981-2021. This discrepancy can be reconciled by the observed positive-to-negative phase shift of Interdecadal Pacific Oscillation (IPO) and the negative-to-positive phase shift of Atlantic Multidecadal Oscillation (AMO), which increase and reduce Arctic ARs over the Atlantic and Pacific sectors, respectively. Removing the influence of the IPO and AMO can reduce the projection uncertainties in near-future Arctic AR trends by about 24%, which is important for constraining projection of Arctic warming and the timing of an ice-free Arctic.

The Arctic has undergone dramatic changes in recent decades, with a warming rate nearly four times faster than the global average[1], a phenomenon known as Arctic Amplification (AA). Concurrent with AA, the extent of Arctic sea ice has shown a substantial decline, with the strongest decline over the western Arctic during summer and over the Barents Sea during winter[2]. While the summer western Arctic sea ice decline has been attributed to the recent persistent positive Pacific North American pattern[3], the strengthening and warming of the Atlantic inflow, which has been termed "Atlantification" of the Arctic Ocean, has warmed the Barents Sea and contributed to the winter sea ice decline there[4,5]. AA and its associated sea ice loss are expected to have profound repercussions on the local human and natural systems[6–12]. Through modulation of large-scale circulations, the influence of AA can be felt beyond the Arctic[13–18]. Several key mechanisms have been identified to contribute to AA, including local feedbacks, such as the ice albedo feedback[19–21], lapse rate feedback[22,23], cloud and

water vapor feedbacks[24,25], and poleward energy transport[26–29]. Among all the poleward energy transport components, atmospheric moisture transport is especially effective in inducing Arctic warming[29].

It has long been known that atmospheric rivers (ARs), long and narrow corridors of intense moisture transport in the atmosphere, are responsible for most of the poleward atmospheric moisture transport over mid-latitudes[30]. Recent studies have further revealed that 70 – 80% of the atmospheric moisture transported into the Arctic is accomplished by ARs[31–35], suggesting their potential contribution to AA. In addition, the intrusion of substantial moisture and heat into the Arctic by ARs can rapidly moisten and warm the Arctic atmosphere and subsequently enhance downward longwave radiation[36–38]. Hence at the synoptic time scale, ARs are efficient drivers of heat extremes and rapid sea ice loss over the Arctic[32,37,39,40].

As intense moisture transport in the atmosphere, ARs can be characterized by both atmospheric moisture content and wind speed.

[1]Atmospheric, Climate, and Earth Sciences Division, Pacific Northwest National Laboratory, Richland, WA, USA. [2]Department of Atmospheric and Oceanic Sciences, University of California Los Angeles, Los Angeles, CA, USA. [3]Department of Atmospheric Sciences, University of Washington, Seattle, WA, USA. [4]Department of Earth and Atmospheric Sciences, Indiana University, Bloomington, IN, USA. [5]Cooperative Institute for Research in Environmental Sciences, University of Colorado, Boulder, CO, USA. [6]National Snow and Ice Data Center, University of Colorado, Boulder, CO, USA. [7]Department of Atmospheric and Oceanic Sciences, University of Colorado, Boulder, CO, USA. [8]Department of Oceanography, Naval Postgraduate School, Monterey, CA, USA. ✉e-mail: weiming.ma@pnnl.gov; Hailong.Wang@pnnl.gov

Given the recent warming in the Northern Hemisphere, more moisture is available to fuel the formation of ARs. It was shown that ARs or extreme moisture intrusions have been increasing over the Atlantic sector of the Arctic during winter[32,37,41]. This increase in ARs contributes to the decline in sea ice over the Barents-Kara Sea[32]. The Arctic-wide annual AR counts have shown an upward trend in the past four decades, with the location of peak AR occurrence frequency shifting poleward from land to the Arctic Ocean[42]. However, a more systematic understanding on the spatial distribution of trends in the Arctic AR occurrence frequency in recent decades and the associated driving mechanisms is still lacking.

In addition to anthropogenic warming, oceanic internal variability also has significant influences on both large-scale circulations and moisture redistribution. In particular, the Interdecadal Pacific Oscillation (IPO) and the Atlantic Multidecadal Oscillation (AMO), the dominant internal modes of variability over the Pacific and North Atlantic, respectively, have been found to exert far-reaching impacts on the regional and global climate[43,44]. Their influences over the Arctic are especially pronounced. Through the modulation of poleward oceanic and atmospheric energy transport, the phase shift of the IPO and AMO can either accelerate or dampen the warming and sea ice loss over the Arctic on multi-decadal time scales[45–47]. As part of the poleward energy transport, ARs likely vary in their strength and occurrence frequency as the IPO and AMO undergo phase shifts.

In this study, we systematically quantify Arctic AR trends in multiple sources of data and investigate the driving mechanisms over the past four decades. We discover that the observed Arctic AR frequency increases faster over the Atlantic sector compared to the Pacific sector, while models simulate a spatially more uniform anthropogenically driven trend. Using ensembles of fully coupled and atmosphere-only model experiments, we reconcile the discrepancy between the observed

AR trends and the model ensemble mean trends using the observed phase shift of IPO and AMO. Given the strong connection between ARs and these two interdecadal modes, improving decadal prediction of the phase shift of IPO and AMO may lead to a better projection of future Arctic AR changes. This in turn can result in better projections of future changes in Arctic extreme weather events, the rate of future Arctic warming, as well as the timing of a sea ice-free Arctic.

## Results

### Observed and simulated historical Arctic AR trends

Although significant warming has been observed over the entire Arctic, this has not translated into a significant Arctic-wide positive trend in AR frequency, which is defined as the fraction of time (in percentage) when AR is detected at a grid point. In the past four decades, significant increases in AR frequency are observed mostly over the Atlantic sector of the Arctic, including the Greenland Sea and Baffin Bay (Fig. 1a). The magnitude of these increases reaches as high as 0.9% decade⁻¹. With the climatology of AR frequency over the Greenland Sea and Baffin Bay being 6-9% and 3-5%, respectively (Supplementary Fig. 1), there have been substantial increases in AR activities over the Atlantic sector. However, as another major pathway for ARs into the Arctic, the Pacific sector experiences a weaker increase in AR frequency, with significant trends confined only to a small region of the Chukchi Sea. Averaging over the Atlantic sector (red box in Fig. 1a) and the Pacific sector (magenta box in Fig. 1a) individually, ARs have been increasing at a rate of about 0.42 (0.49) % decade⁻¹ and 0.19 (0.29) % decade⁻¹ over the respective regions in ERA5 (MERRA-2) (Fig. 1e). The occurrence frequency of ARs over the Atlantic sector has thus increased about twice as fast as those over the Pacific sector.

Decomposing the observed trend into a dynamical component (Fig. 1b), driven by changes in atmospheric circulation, and a

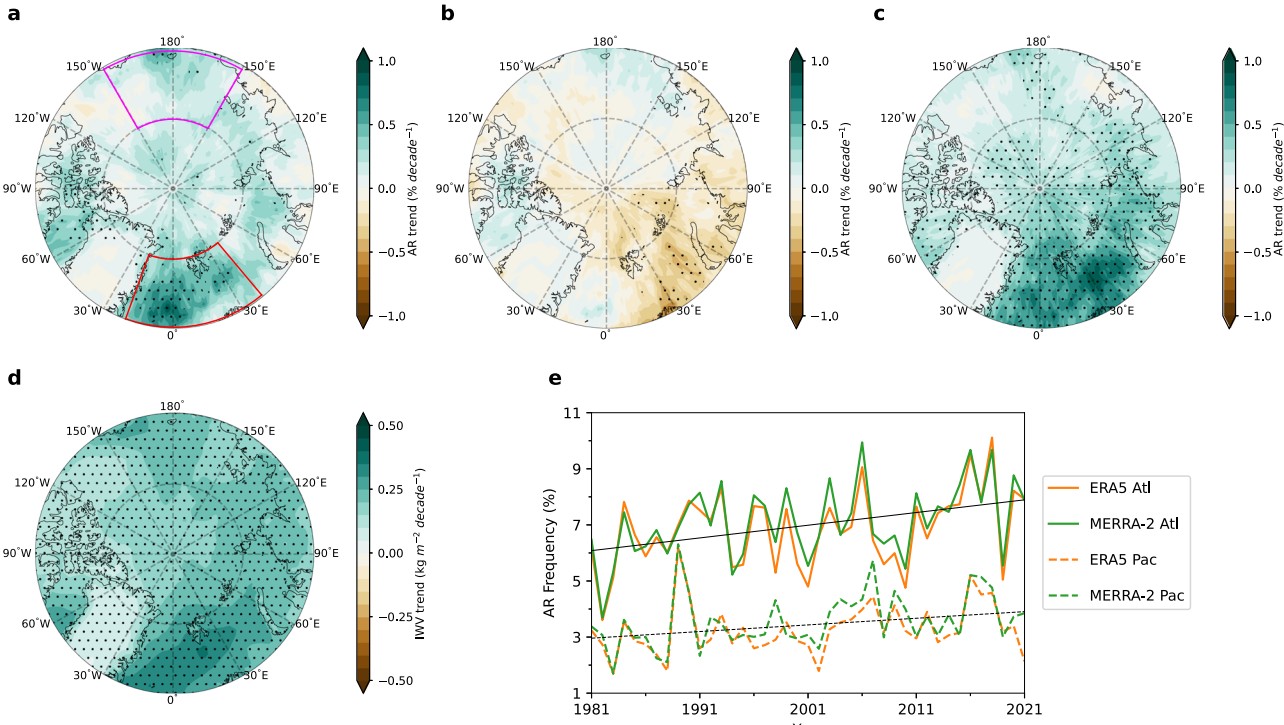

**Fig. 1 | Observed Arctic atmospheric river (AR) trends. a** Arctic AR frequency trend during 1981–2021 in ERA5. The decomposed dynamical and thermo-dynamical contributions are shown in (**b**) and (**c**), respectively. **d** The observed trend in column-integrated water vapor (IWV) during 1981–2021 in ERA5. Stippled areas in (**a**), (**b**), (**c**), and (**d**) indicate significant trends at the 0.05 level based on the Student's t-test. **e** Time series of area-average AR frequency over the Atlantic sector (solid lines; area outlined by the red box in (**a**) and Pacific sector (dashed lines; area outlined by the magenta box in (**a**). The black lines in (**e**) are the mean linear trends of the two reanalysis datasets (ERA5 and MERRA-2). The ensemble mean trends over both the Atlantic sector and Pacific sector are significant at the 0.05 level based on the Student's t-test.

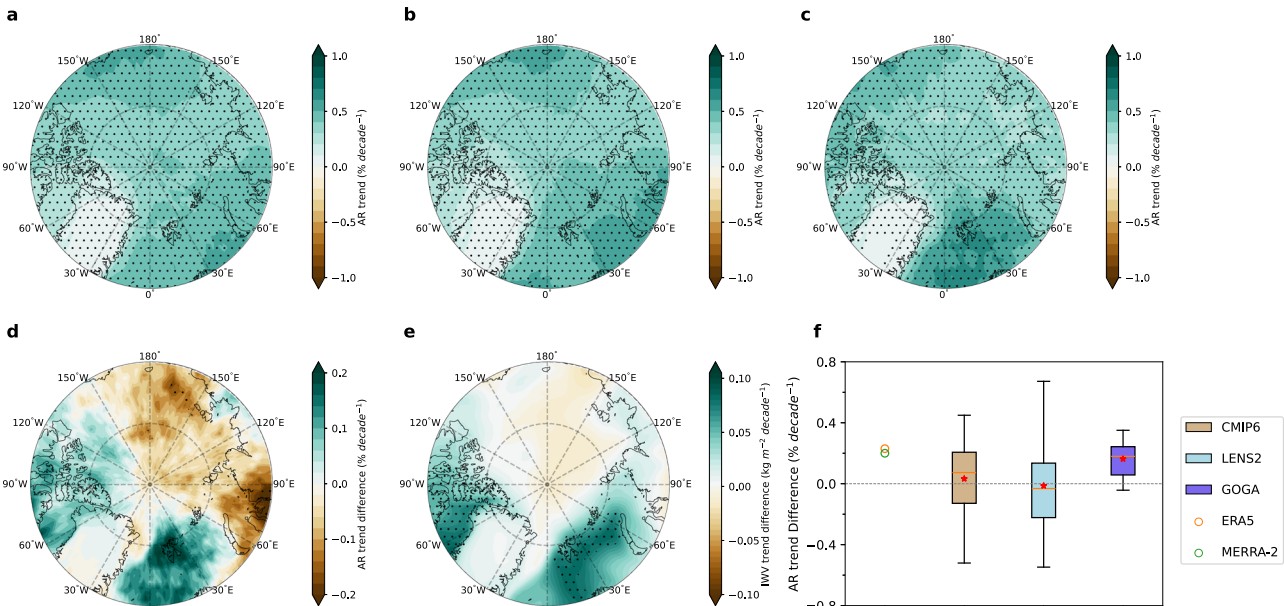

**Fig. 2 | Simulated Arctic atmospheric river (AR) trends during 1981-2021.**
**a** Ensemble mean trend in the Arctic AR frequency simulated in LENS2. **b** Ensemble mean Arctic AR frequency trend in LENS2 due to thermodynamical changes. **c** Ensemble mean Arctic AR frequency trend in GOGA. **d** Difference between the ensemble mean AR frequency trends in GOGA and LENS2 (GOGA – LENS2). **e** Same as (**d**) but for the column-integrated water vapor (IWV). The stippled areas in (**a**),

(**b**), (**c**), (**d**), and (**e**) indicate trends or differences that are significant at the 0.05 level based on the Student's t-test. **f**, Trend differences between the Atlantic sector (red box in Fig. 1a) and Pacific sector (magenta box in Fig. 1a) in reanalyses (circles) and simulations (bars and whiskers). The orange lines (red stars) in (**f**) represent the ensemble median (mean). The boxes represent the 25th–75th percentile range of the spread. The whiskers denote the maximum and minimum of the spread.

thermodynamical component (Fig. 1c) associated with changes in moisture, reveals that the faster increases in AR frequency over the Atlantic sector are due to a stronger atmospheric moistening over the region (Fig. 1d; Methods). This stronger moistening over the Atlantic sector intensifies both the mean and extreme integrated water vapor transport (IVT) trends there and results in more frequent AR occurrence (Supplementary Fig. 2). Changes in circulation tend to partly offset the positive AR trend over the Atlantic sector. Examining the dynamical trends in individual seasons further reveals that the negative annual dynamical contribution over the Atlantic sector is dominated by the negative dynamical trends in winter and summer (Supplementary Fig. 3). It is also worth mentioning that the observed changes in AR frequency shown in Fig. 1 do not depend on the dataset used, as we also see such changes in MERRA-2, although the trend magnitude in MERRA-2 tends to be slightly stronger (Supplementary Fig. 4). Furthermore, the spatial pattern in the Arctic AR frequency trend shown in this study is not sensitive to the use of different AR detection algorithms (Supplementary Fig. 5). This observed pattern can also be identified in the AR datasets derived using various global AR detection algorithms that participated in ARTMIP[48], except for the ones that detect almost no AR over the Arctic (Supplementary Fig. 6).

The observed historical trends in Arctic AR frequency can be driven by two factors: (1) anthropogenic forcing and (2) interdecadal internal variability. Based on the CESM2 Large Ensemble (LENS2) (Fig. 2), the CMIP6 multi-model ensemble (Fig. 2e and Supplementary Fig. 7) and two other single model ensembles (Supplementary Fig. 11a, c), anthropogenic forcing alone leads to a uniform increase in AR frequency over the entire Arctic, resulting mostly from the moistening of the atmosphere (Fig. 2a, b), while the contribution from circulation changes is negligible (Supplementary Fig. 8a). Such discrepancies between the observed and the simulated trends suggest that factors, such as the observed internal variability and/or model deficiency in capturing the forced response, likely play a role in shaping the spatially differing observed trend. To elucidate whether the internal variability associated with the observed sea surface

temperature (SST) and sea ice variability have contributed to the observed AR trend, we employ a 10-member atmosphere-only CESM2 ensemble driven by the observed SST/sea ice, called GOGA (Global Ocean Global Atmosphere) experiments. Compared with the anthropogenically driven trend in LENS2, GOGA successfully reproduces the observed stronger AR trend over the Atlantic sector and weaker trend over the Pacific sector, driven also by a faster moistening of the atmosphere over the Atlantic sector (Fig. 2c–e and Supplementary Fig. 8c). GOGA also simulates a negative AR trend over the Atlantic sector due to dynamical changes (Supplementary Fig. 8b), which is consistent with observations (Fig. 1b). Furthermore, the trend differences between the Atlantic sector and Pacific sector in observations fall within the 25th – 75th percentile range of the intermember spread in GOGA, but outside of that range in LENS2, of which the ensemble mean trend difference is very close to zero (Fig. 2f). Thus the historical SST/sea ice variability is key to understanding the observed pattern in Arctic AR trends.

**Interdecadal Arctic AR trend modulated by the IPO and AMO**
The above analyses suggest that large-scale SST patterns play an important role in modulating Arctic AR trends at the interdecadal time scale. To better understand how internal variability associated with large-scale SST patterns influences interdecadal Arctic AR trends, maximum covariance analysis (MCA) is applied to the covariance matrix between the internally generated Arctic AR trends and the internally generated global SST trends (60°S–70°N) across all 50 members of the LENS2. The first two modes of MCA account for the majority of the covariance between the two fields. The first mode, which explains about 65% of the covariance, exhibits strong increases in ARs over most of the Arctic, especially over the Atlantic sector (Fig. 3a). The corresponding spatial pattern of SST trends shows a positive IPO over the Pacific and a basin-wide warming over the North Atlantic (Fig. 3b). Similar to the observed Arctic AR trends, the second mode, which accounts for about 12% of the covariance, displays a dipole pattern with an increase in AR frequency over the Atlantic

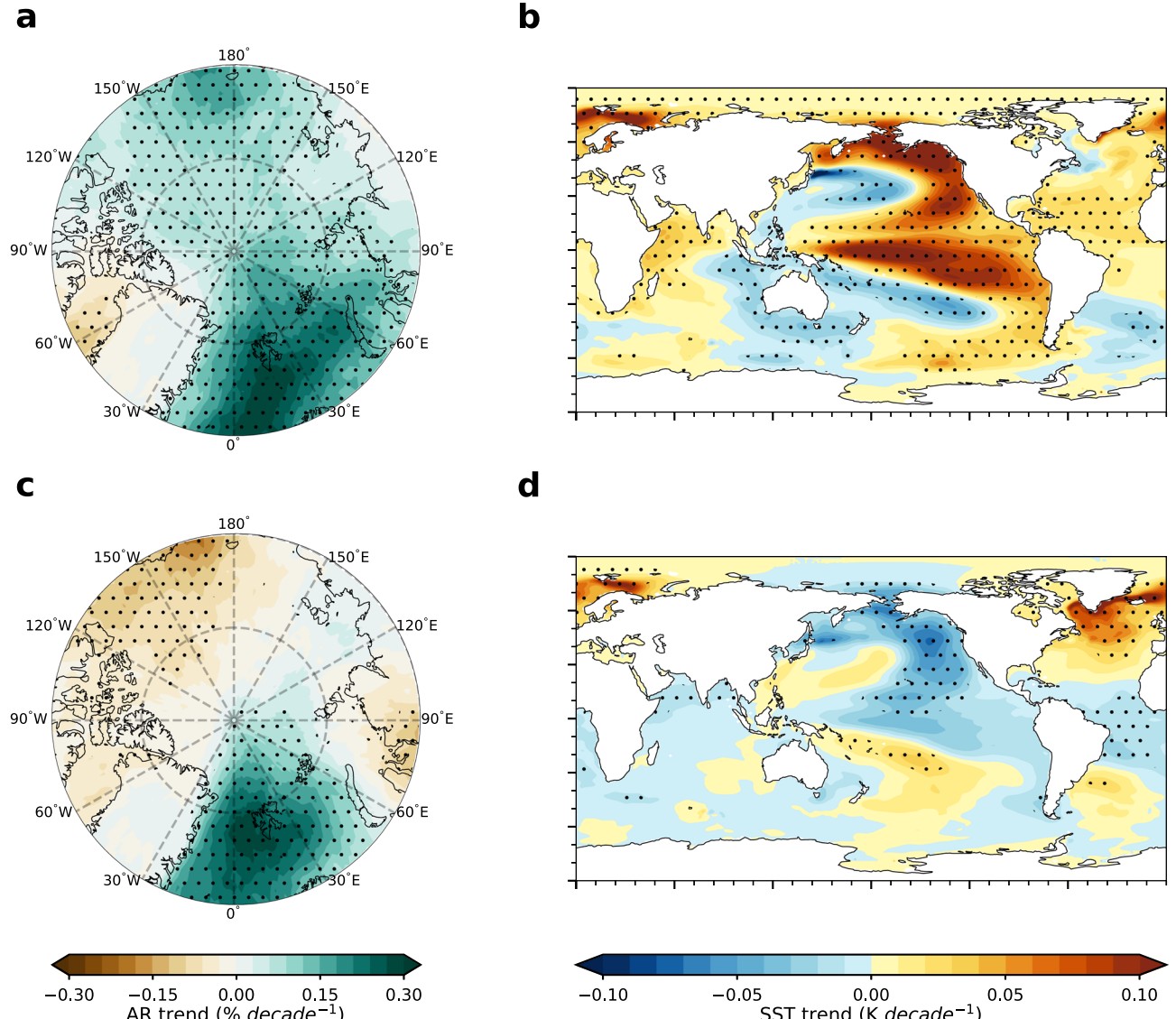

**Fig. 3 | Leading modes of covarying Arctic atmospheric river (AR) trends and global sea surface temperature (SST) trends due to internal variability. a** Spatial pattern of the Arctic AR frequency trend associated with the first mode of maximum covariance analysis (MCA). To focus on the portion of the co-variability between Arctic AR trend and global SST trend driven by internal variability, the ensemble mean trends of both the Arctic AR frequency and SST in the 50-member LENS2 (Fig. 2a) have been removed from individual members before applying the MCA. **b** SST trend pattern associated with the first mode. **c, d** Same as (**a**), (**b**) but for the second mode of MCA. The fraction of covariance explained by the first and second mode is 65% and 12%, respectively. These patterns are obtained by regressing the internally generated trends across 50 members onto their respective standardized expansion coefficients. Stippled areas indicate that regressions are significant at the 0.05 level based on the Student's t-test.

sector and a decrease in the Pacific sector (Fig. 3c). The corresponding spatial pattern of SST trends shows a positive AMO over the North Atlantic and a negative IPO-like pattern over the Pacific (Fig. 3d). This second mode resembles the observed IPO, which has shown an overall negative phase shift during the past four decades, while the observed AMO exhibits a positive phase shift (Supplementary Fig. 9). The results of the second mode indicate that the observed phase shift of the IPO and AMO favors the increase and reduction in ARs over the Atlantic and Pacific sectors, respectively.

Based on the MCA, it is clear that the phase shift of the IPO and AMO exerts a strong control on the interdecadal Arctic AR trends. To further demonstrate the tight relationship between these two modes and the Arctic AR trend, we performed an intermember regression of SST trends onto the Arctic spatial mean AR trends based on LENS2. The regression features an SST pattern that shows a marked similarity to the positive IPO over Pacific and positive AMO over the North Atlantic

(Fig. 4a). Furthermore, the Arctic mean AR trends exhibit a significant positive correlation with both the IPO and AMO trends (Fig. 4b, c). Similar relationships also can be found between the mean AR trends in the Pacific sector (magenta box in Fig. 1a) and the IPO trends, as well as between the mean AR trends in the Atlantic sector (red box in Fig. 1a) and the AMO trends (Supplementary Fig. 10). However, the correlation for the former weakens and for the latter strengthens. This weakened correlation between the mean AR trends in the Pacific sector and the IPO trends is partly caused by an outlier member with a slightly negative AR trend over the Pacific sector. Removing this member can increase the correlation to about 0.3, which is significant at the 0.05 level. It may also be attributed to the relatively smaller area extent of the Pacific sector than the entire Arctic. When transitioning the regional focus from the entire Arctic to the Pacific sector alone, other internal variability processes, such as atmospheric internal variability[3,49], likely play increasingly more important roles in

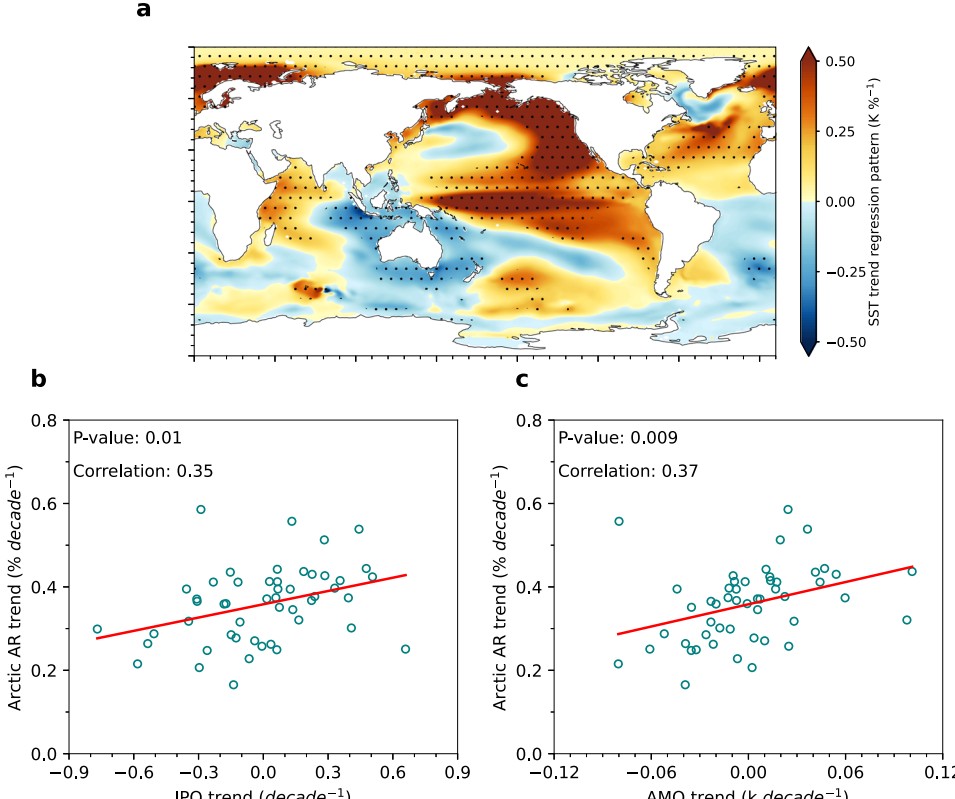

**Fig. 4 | Relationship between the Arctic spatial mean atmospheric river (AR) frequency trends and the Interdecadal Pacific Oscillation (IPO)/Atlantic Multidecadal Oscillation (AMO).** **a** LENS2 intermember regression of the sea surface temperature (SST) trends onto the Arctic spatial mean AR frequency trends. The regression pattern is obtained by regressing the SST trends at each grid point across all 50 members onto the spatially averaged Arctic AR frequency trends across all 50 members. Stippled areas indicate that the regression is significant at the 0.05 level based on the Student's t-test. **b** shows scatterplots between the Arctic mean AR frequency trends and the IPO trends, where the red lines show the regression of data points for 50 members of LENS2. **c** same as in (**b**), but for the AR trends and AMO trends.

modulating the interdecadal variability of ARs over this specific region. The relative roles of the IPO versus other atmospheric internal variability in modulating the AR variability over the Pacific sector at different time scales warrant further studies. Nevertheless, these results suggest that a concurrent negative-to-positive phase shift of the IPO and AMO likely enhances Arctic-wide AR activities, and is consistent with the effects of the interdecadal modes on the amplified early 20th century Arctic warming[47]. The intermember regression pattern between the Arctic AR trends and SST trends is not unique to the LENS2. Similar regression patterns can also be identified in two other large ensembles based on different climate models (Supplementary Fig. 11b, d), further confirming the robust impacts of the IPO/AMO on ARs in the Arctic.

### Mechanisms of the IPO and AMO in driving Arctic AR trends

To investigate how the IPO and AMO influence the Arctic AR trends, we regress the AR frequency variability onto the IPO and AMO index, respectively, in the LENS2. As shown in Fig. 5a, the positive IPO drives a strong increase in AR frequency over the Pacific sector, and a slightly reduced AR frequency over the Atlantic sector, including the Baffin Bay and Barents-Kara Sea. On the other hand, the positive AMO leads to widespread increases in ARs over most of the Arctic, especially over the Greenland Sea, while AR frequency only decreases over a confined area of the Pacific sector (Fig. 5d). The results here further suggest that the present combination of observed negative IPO and positive AMO favors increasing ARs over the Atlantic sector and decreasing ARs over the Pacific sector.

The phase shift of the IPO and AMO can modify both atmospheric circulation and moisture fields. To understand how the IPO- and AMO-

related changes in circulation and moisture modulate AR frequency, we regress the AR frequency variability due only to the circulation variability onto the IPO and AMO index to obtain the influence of circulation changes (Methods). The residual of the regression is treated as the influence of the IPO and AMO on the AR variability due to the moisture changes. By moistening the Arctic atmosphere, the positive IPO and AMO increase AR frequency over most of the Arctic, especially over their respective sectors (Fig. 5c, f). In contrast, the associated circulation changes tend to drive regional decrease of AR frequency (Fig. 5b, e). Specifically, a positive IPO induces a positive and a negative sea level pressure (SLP) anomaly over the Beaufort Sea and Northwest Eurasia, respectively (Supplementary Fig. 12a). These SLP anomalies induce northwesterly and northeasterly surface wind anomalies over the Beaufort Sea and Barents-Kara Seas, respectively. Since Arctic ARs are usually associated with southerly wind, the circulation anomalies over the Beaufort Sea and Barents-Kara Seas thus act to reduce AR activities there. In response to the positive AMO, high SLP anomalies form over almost the entire Arctic, with negative SLP anomalies found over mid-latitude regions. This SLP anomaly pattern resembles the negative phase of Arctic Oscillation[50]. The high SLP anomalies have two centers, including one located over the Laptev Sea and the other south of Iceland. The high SLP anomaly over the Laptev Sea extends eastward into the Beaufort Sea and induces northeasterly wind anomaly there (Supplementary Fig. 12c). A positive AMO thus acts to reduce ARs over the Beaufort Sea. Over the North Atlantic, the high SLP anomaly south of Iceland is accompanied by a low SLP anomaly further south. Consistent with previous studies[51–53] which show that a positive AMO can induce a negative North Atlantic Oscillation (NAO), especially during the cold season, this dipole SLP anomaly pattern projects onto the

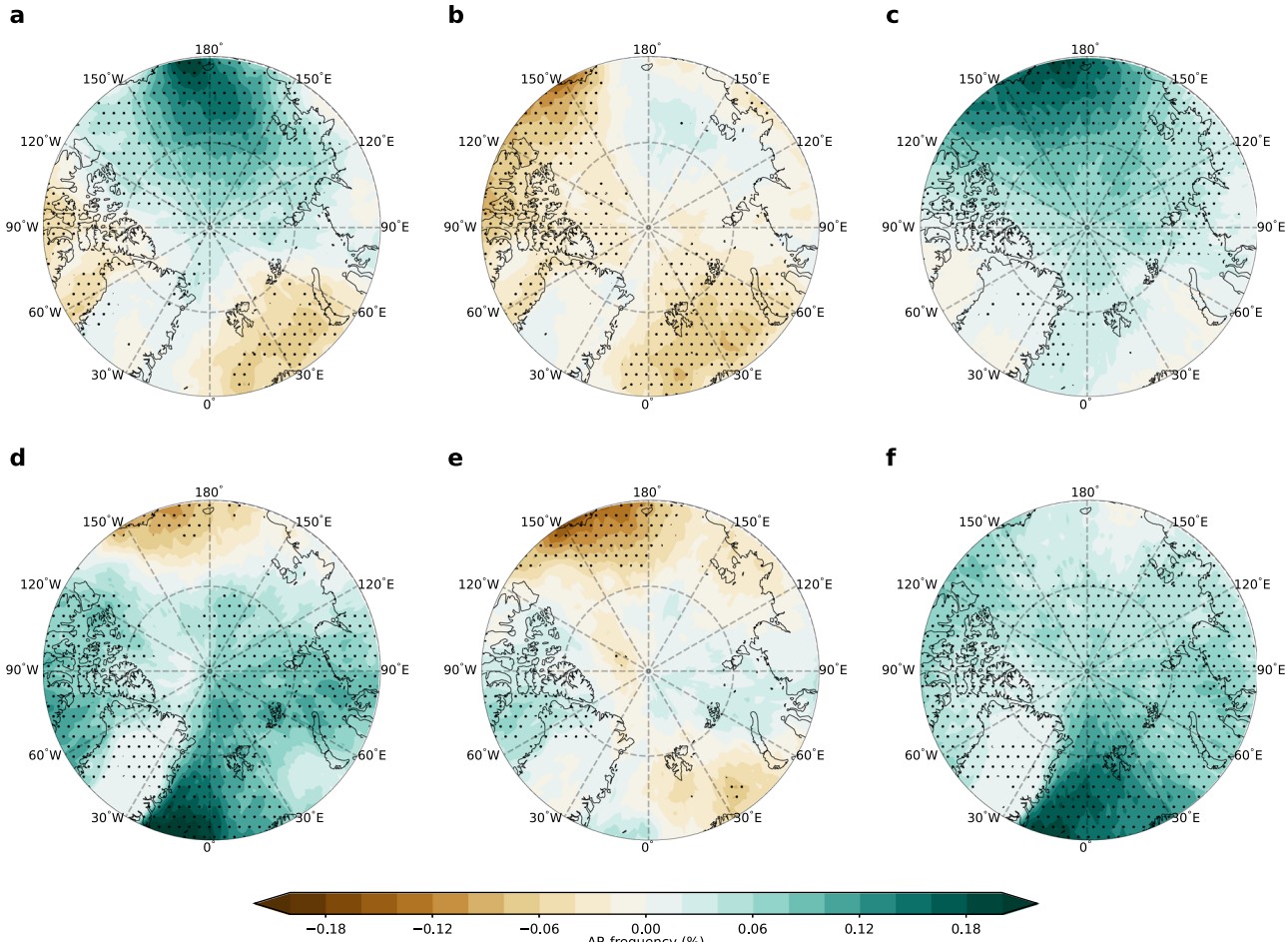

**Fig. 5 | Mechanisms of the Interdecadal Pacific Oscillation (IPO) and Atlantic Multidecadal Oscillation (AMO) in driving Arctic atmospheric river (AR) trends. a** Ensemble mean pattern of the regression of Arctic AR frequency time series onto the standardized IPO index, which represents the total effect of IPO on AR frequency trend. **b** Similar to (**a**), but obtained by regressing the Arctic AR frequency time series associated with circulation variability onto the standardized IPO index, which represents the dynamical contribution of IPO to the total AR frequency trend. **c** Obtained by taking the difference between (**a**) and (**b**), representing the thermodynamical contribution of the IPO to the total AR frequency trend. **d**–**f** Same as (**a**)–(**c**), but for the AMO. These regression patterns are based on the historical + SSP370 data in LENS2 from 1979 to 2100. Stippled areas indicate the regression is significant at the 0.05 level based on the Student's t-test.

negative phase of NAO[54]. The negative NAO pattern enhances AR activities over the Baffin Bay and suppresses those over the Barents Sea (Fig. 5e), in line with the role of NAO in modulating poleward moisture transport[35,55]. Although the results presented here are based only on the LENS2, such IPO and AMO associated circulation changes are consistent with observations[56–58], and can be reproduced by two other large ensembles, except for the AMO-induced circulation anomalies in CNRM (Supplementary Fig. 13). These findings thus further support the strong link between the observed phase shift of IPO/AMO and the observed Arctic AR trends.

### Constrained projection of near-future Arctic ARs with IPO and AMO

Given the strong influence of IPO and AMO on the interdecadal Arctic AR variability, both IPO and AMO can potentially be used to constrain Arctic AR projections. Under the SSP370 warming scenario, Arctic AR frequency is projected to increase at an even faster rate compared to the historical period, due to the enhanced moistening of the Arctic atmosphere (Fig. 6a). However, there is a large spread across all 50 members in the spatially averaged Arctic AR trends, ranging from -0.2 to -0.7% decade$^{-1}$ (Fig. 6b). To demonstrate how the IPO and AMO serve as a constraint for Arctic AR projections, we exclude their influence from each member by removing the Arctic AR variations that are linearly associated with the IPO and AMO, separately, from each

member (Methods). After the influence of the IPO and AMO is removed, the spread across all members, as measured by the standard deviation, reduces considerably from 0.11 to 0.084 % decade$^{-1}$, a 24% reduction in the spread or uncertainty (Fig. 6b). Further analyses reveal that the reduction in uncertainty is mostly contributed by the removal of the AMO impact, which alone can lead to a 23.4% reduction in uncertainty, while the contribution from IPO removal is minor (1.5%) (Supplementary Fig. 14). Such a minor contribution from the IPO is possibly due to the enhanced future role of the forced AR trend over the Pacific sector (Fig. 6a). That is, the strong forced trend in the Pacific sector can overwhelm the signal from the IPO and weakens the IPO's influence over the region. In contrast, the near-future forced AR trends over the Greenland Sea, where AMO exerts a strong influence, weaken slightly compared to the historical trend (Fig. 6a vs. Fig. 2a). This makes the role of the AMO even more important under the enhanced warming. Nevertheless, the results here suggest that a better prediction of the future evolution of the IPO and AMO increases the confidence in AR projection over the Arctic.

## Discussion

In this study, we show that Arctic ARs over the Atlantic sector have been increasing about twice as fast as those over the Pacific sector during 1981–2021. This uneven increase in ARs is driven by a greater moistening of the atmosphere over the Atlantic sector. On the other

**a**

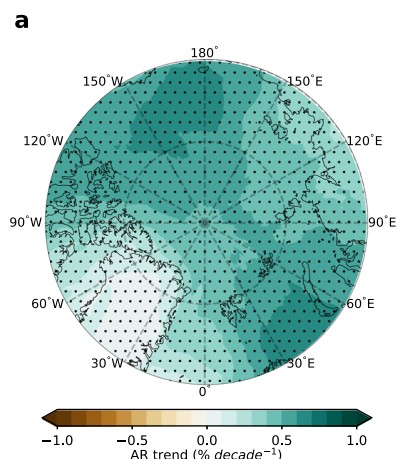

**b**

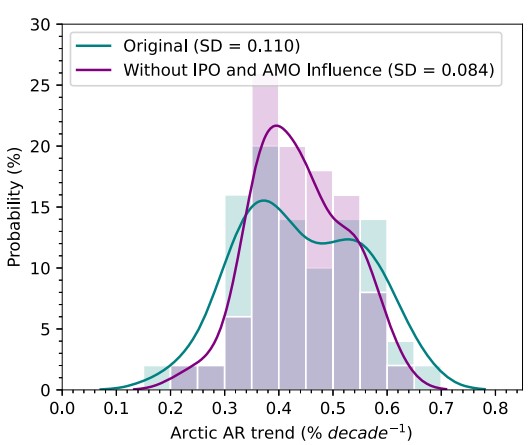

**Fig. 6 | Near-future Arctic atmospheric river (AR) frequency trends with and without the influences of the Interdecadal Pacific Oscillation (IPO)/Atlantic Multidecadal Oscillation (AMO). a** Ensemble mean near-future (2024-2064) Arctic AR frequency trend in LENS2. Stippled areas indicate the trend is significant at the 0.05 level based on the Student's t-test. **b** Histograms (bars) and the probability density function based on kernel density estimation (lines) of the near-future Arctic spatial mean AR frequency trends. The teal bars and line represent the original Arctic mean AR frequency trends, while the purple bars and line denote the Arctic mean AR frequency trends without the influences of the IPO and AMO. Note that the gray bars indicate an overlap between purple and teal.

hand, simulations from the 50-member coupled CESM2 large ensemble (LENS2), the CMIP6 multi-model ensemble, the ACCESS and the CNRM ensembles all show that, in climate models, anthropogenic forcing alone leads to a uniform increase in ARs over the Arctic. The discrepancy between the observed and model ensemble mean trends can be reconciled by accounting for the observed SST and sea ice in the model experiments. We further found that the observed negative and positive phases of the IPO and AMO are critical to explaining the observed Arctic AR trends. Both the IPO negative phase and AMO positive phase favor an increase in ARs over the Atlantic sector and a decrease over the Pacific sector through their influence on atmospheric moisture and large-scale circulation. Under global warming, Arctic ARs are expected to increase in an even faster pace, mostly driven by the increasing atmospheric moisture. Using the 50-member LENS2 as an example, we further demonstrate that removing the influence of the IPO and AMO from the projected changes in the Arctic ARs can reduce the projection uncertainty by about 24%. Given the strong coupling between Arctic AR variability and the IPO/AMO, improving the prediction of future IPO and AMO evolution is expected to lead to an improved projection of future Arctic ARs and thus their impact on sea ice and AA.

## Methods
### Observational datasets
Two different reanalysis products are employed in this study to cross-validate the robustness of results. The results shown in the main text are based on the European Centre for Medium-Range Weather Forecasts Reanalysis, version 5 (ERA5)[59], and the results in Supplementary Fig. 4 are based on the Modern-Era Retrospective analysis for Research and Applications, version 2 (MERRA-2)[60]. Both ERA5 and MERRA-2 are regridded to a spatial resolution of $1° \times 1°$ before conducting the analysis. We focus on the period of 1980−2021. Results are based on daily data that are obtained by averaging 6-hourly data at 00, 06, 12, and 18 UTC. The observed SST is based on the Met Office Hadley Centre's sea surface temperature dataset (HadSST)[61].

### Model simulations
To disentangle the roles of anthropogenic forcing versus internal variability in driving the observed Arctic AR trends, we use two sets of large ensemble simulations from the Community Earth System Model, version 2 (CESM2). To quantify the influence of anthropogenic forcing,

we employed the fully coupled 50-member CESM2 large ensemble (LENS2)[62], which can simulate a Arctic AR climatology comparable to observations (Supplementary Fig. 1b). LENS2 is driven by the historical forcing from 1850 to 2014, and SSP370 forcing afterwards. LENS2 consists of 100 members in total and can be further divided into two 50-member sub-ensembles. These two sub-ensembles differ only in the biomass burning (BMB) aerosol forcing, with one driven by the CMIP6 BMB, and the other driven by the smoothed CMIP6 BMB. Other than that, they are identical in forcing. CMIP6 BMB utilizes satellite-based estimation of aerosol emission data during 1997-2014, which give higher interannual variability compared to the data before and after that time. It has been found that Arctic climate in the fully coupled CESM2 shows high sensitivity to such enhanced variability in the original CMIP6 BMB forcing, leading to too strong Arctic sea ice loss and warming in the early 21st century[63]. However, these spurious trends in Arctic sea ice and temperature vanish when the model is driven by a smoothed CMIP6 BMB, making the historical climate more comparable to observations. Despite being able to simulate a uniform positive AR trend over the Arctic, we indeed found that the ensemble mean trend in the sub-ensemble driven by CMIP6 BMB is stronger than the trend in the sub-ensemble driven by the smoothed BMB (not shown). In addition, the magnitude of the trend in the sub-ensemble driven by the smoothed CMIP6 BMB is also more comparable to the ensemble mean trend in GOGA. Considering these findings and the goal of this study, we decided to adopt the 50-member sub-ensemble driven by the smoothed BMB emission data.

To identify the roles of the observed SST and sea ice in shaping the observed Arctic AR trends, we also looked at a 10-member atmosphere-only ensemble from the same CESM2 model. This ensemble is driven by observed SST from NOAA Extended Reconstruction Sea Surface Temperature Version 5 (ERSSTv5) and sea ice from Hadley Centre sea ice (HadISST1) from 1880 to 2021, termed Global Ocean Global Atmosphere (GOGA). The atmospheric forcings of GOGA are nearly identical to those in LENS2, except that GOGA is driven by the CMIP6 BMB. Because SST and sea ice over the Arctic are prescribed in GOGA, the high sensitivity of the Arctic climate to CMIP6 BMB found in LENS2 is muted in GOGA. Since LENS2 and GOGA are driven by nearly identical forcings and based on the exact same model, the differences between the ensemble mean trends of these two ensembles can thus be treated as the influence of the observed SST and sea ice.

To further test whether the results of LENS2 are robust, three additional large ensembles are also analyzed in this study. The first ensemble consists of 23 coupled models from CMIP6 (CMIP6 ensemble; see table S1 for model information). Many of these models have more than one ensemble member. Only members with the same variant index "r1i1p1f1" are included in this ensemble. The second ensemble is a 40-member coupled ensemble based on the model ACCESS-ESM1-5 (ACCESS ensemble). The third ensemble is a 30-member coupled ensemble based on the model CNRM-CM6-1 (CNRM ensemble). We focus on the period from 1981 to 2021 for both the CMIP6 and ACCESS ensembles, but 1979 to 2014 for the CNRM ensemble because data under SSP370 forcing is not provided for this ensemble. All three ensembles are driven by the CMIP6 historical forcing up until 2014, and under SSP370 forcing afterwards for the CMIP6 and ACCESS ensembles. Daily outputs are used for all model simulations.

## AR detection algorithm

The AR detection algorithm used in this study is based on the integrated water vapor transport (IVT) developed in ref. 64 with minor modification for the Arctic application. This algorithm is an updated version of the algorithm originally introduced in ref. 65, which is widely used in the AR research community and also recommended by the Atmospheric River Tracking Method Intercomparison Project (ART-MIP) for AR studies over polar regions[66]. Common criteria shared by both algorithms include: (1) a monthly dependent $85^{th}$ percentile of the IVT magnitude or 100 kg m$^{-1}$ s$^{-1}$, whichever is larger, is used as the threshold to identify contiguous regions of enhanced IVT ("object"); (2) the mean meridional (poleward) IVT of the "object" needs to be greater than 50 kg m$^{-1}$ s$^{-1}$; (3) more than half of the grid points of the "object" have an IVT direction within 45° from the "object" mean IVT; (4) the "object" is longer than 2000 km, with an length-to-width ratio greater than two. Compared to the original algorithm, major refinements on the updated algorithm include: (1) iterative thresholds are enabled to increase the chance of an "object" to be detected as AR; (2) improvements on the identification of the AR axis which lead to better characterization of the AR length and orientation; (3) tracking of individual ARs across space and time. Readers are referred to refs. 64,65 for more detailed descriptions of the algorithm. Since our focus is on the Arctic ARs, which are usually near the end of their life cycle, according to ref. 55, we relax the length requirement from 2000 km to 1500 km. In addition, for computational efficiency, iterative thresholds are not implemented in this study. The AR statistics based on the algorithm used in this study are thus nearly identical to those based on the original algorithm developed in ref. 65, which has been confirmed from the AR results.

IVT for observations, LENS2 and GOGA model ensembles is calculated as

$$\text{IVT} = \sqrt{(\text{IWV*}\mathbf{U850})^2 + (\text{IWV*}\mathbf{V850})^2} \qquad (1)$$

where IWV is the column-integrated water vapor, U850 and V850 are the zonal and meridional wind at 850 mb, respectively. Because IWV is not available in the CMIP6, ACCESS and CNRM ensembles, IVT in these three ensembles is calculated by vertically integrating the moisture flux at 1000, 850, 700, and 500 mb following:

$$\text{IVT} = \frac{1}{g} \int_{1000}^{500} \mathbf{u} q \, \mathrm{d}p \qquad (2)$$

where $g$ is the gravitational acceleration, $\mathbf{u}$ is the horizontal winds and $q$ is specific humidity. The reason why we calculated IVT in observations, LENS2 and GOGA differently is because the required data at 1000 mb and 700 mb are not available in GOGA. To facilitate fair comparison among the results in observations, LENS2 and GOGA that

we present in the main text, only IWV and winds at 850 mb are used to calculate IVT. However, we have confirmed that the AR statistics based on IVT calculated using Eq. (1) is very similar to those based on Eq. (2) in the observational data, except over Greenland where slightly more frequent ARs tend to be detected when using IVT based on Eq. (2).

## Decomposition of dynamical versus thermodynamical contribution to ARs

ARs can be characterized by both moisture and winds. The variability of ARs across different time scales can thus be driven by variability in the moisture field (thermodynamical contribution) and in the wind field (dynamical contribution). To separate the dynamical versus thermodynamical contribution to the interdecadal Arctic AR trend, a scaling method, which was originally developed in ref. 67, is used. To estimate the dynamical contribution, the moisture field is scaled by a scaling factor $\frac{Q_c}{Q_s}$, where $Q_c$ is the seasonal climatological moisture field in each grid point (and at vertical levels for CMIP6, ACCESS and CNRM ensembles) where this scaling factor is applied. $Q_s$ is the seasonal mean moisture field in the same grid point (and vertical levels for CMIP6, ACCESS and CNRM ensembles) for the same season in a particular year. This scaling method is applied to each season separately. We focus on the winter (December, January, and February), Spring (March, April, and May), Summer (June, July, and August) and Fall (September, October, and November). The first DJF starts from the December of 1980 in all datasets, except the CNRM ensemble. The results of the historical climate presented thus start from 1981. The scaled moisture field is then combined with the wind fields to obtain a scaled IVT. By applying this scaling method, we remove the interannual variability of the moisture field from the scaled IVT. The variability in the AR statistics based on the scaled IVT and the IVT threshold derived from the original IVT can thus be treated as the variability due only to the dynamical effect. Similar scaling method can be applied to obtain the thermodynamical contribution directly. However, previous study has found that the two components are largely linearly additive[16]. The thermodynamical contribution is thus indirectly estimated by taking the difference between the total trend and the trend attributed to dynamical changes.

## IPO and AMO definition and their contribution to the uncertainty in near-future Arctic AR trends

Following ref. 47, we define the IPO index as the 7-year running average of the principal component of the first empirical orthogonal function (EOF) for detrended SST anomalies over the Pacific (120°E–70°W, 50°S–60°N). The AMO index is defined as the 7-year running average of the detrended SST anomaly averaged over the North Atlantic (60°W–0°, equator–70°N). In observational data, the detrending is done by removing the linear trend based on the entire period from 1850 to 2021 covered by the HadSST dataset. In simulations, the detrending is done by removing the ensemble mean time series (forced trend) from individual members.

Following refs. 68,69, the AR variability in individual ensemble member $i$ can be expressed as:

$$\text{AR}(i,t) = \text{r}(i)_{\text{AR,IPO}}\text{IPO}(i,t) + \text{r}(i)_{\text{AR,AMO}}\text{AMO}(i,t) + \text{AR}_{\text{res}}(i,t) \qquad (3)$$

where $t$ is time in a year, $\text{r}(i)_{\text{AR,IPO}}(\text{r}(i)_{\text{AR,AMO}})$ is the regression coefficient of the 7-year running average of the detrended AR frequency time series with respect to the standardized IPO (AMO) index for member $i$ during 1979–2100. Equation (3) states that the total AR variability is the sum of three components: (1) $\text{r}(i)_{\text{AR,IPO}}\text{IPO}(i,t)$, the component linearly associated with the IPO index; (2) $\text{r}(i)_{\text{AR,AMO}}\text{AMO}(i,t)$, the component linearly associated with AMO index; (3) $\text{AR}_{\text{res}}(i,t)$, the residual. Based on Eq. (3), the standard deviation (STD) of $\text{AR}(i,t)$ trend distribution across all members can be compared to the STD of $\text{AR}_{\text{res}}(i,t)$ trend distribution across all

members to estimate the contribution of IPO and AMO to uncertainty in the near-future Arctic AR trends. Similarly, individual contributions of IPO and AMO to the uncertainty of near-future Arctic AR trend can be estimated by comparing the STD of $AR(i,t)$ to the STD of $(r(i)_{AR,AMO}AMO(i,t) + AR_{res}(i,t))$, and the STD of $AR(i,t)$ to the STD of $(r(i)_{AR,IPO}IPO(i,t) + AR_{res}(i,t))$, respectively.

## Data availability

ERA5 and MERRA-2 are available at https://cds.climate.copernicus.eu/#!/home and https://gmao.gsfc.nasa.gov/reanalysis/MERRA-2/data_access/. HadSST can be found at https://www.metoffice.gov.uk/hadobs/hadsst4/. LENS2 is available at https://www.cesm.ucar.edu/community-projects/lens2. GOGA can be found at https://www.cesm.ucar.edu/working-groups/climate/simulations/cam6-prescribed-sst. Data from the CMIP6 multi-model ensemble, ACCESS ensemble and CNRM ensemble can be downloaded at https://esgf-node.llnl.gov/search/cmip6/. ARTMIP data can be accessed at https://www.earthsystemgrid.org/dataset/ucar.cgd.artmip.tier1.catalogues.html. The data[70] used to reproduce Figs. 1–6 are available via figshare at https://doi.org/10.6084/m9.figshare.24905301.v1.

## Code availability

The code[71] for the AR detection algorithm used in this study can be downloaded at https://doi.org/10.25346/S6/B89KXF. All code necessary to reproduce the presented results will be available upon request from the corresponding author Weiming Ma (Weiming.ma@pnnl.gov).

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

## Acknowledgements

This research was supported by the U.S. Department of Energy (DOE), Office of Science, Office of Biological and Environmental Research, Regional and Global Model Analysis program area. This research used resources of the National Energy Research Scientific Computing Center (NERSC), a U.S. DOE Office of Science User Facility operated under Contract No. DE-AC02-05CH11231, as part of NERSC award m1199. The Pacific Northwest National Laboratory (PNNL) is operated for DOE by Battelle Memorial Institute under contract DE-AC05-76RLO1830. G.C. is supported by the U.S. NSF grant AGS-2232581 and NASA grant 80NSSC21K1522. Support for B.K. is provided in part by the National Science Foundation through agreement SES–1754740 and the Indiana University Environmental Resilience Institute. We thank Travis O'Brien for discussion on AR detection and sharing the ARTMIP data, and thank Rudong Zhang for contribution to the precedent analysis of Arctic ARs.

## Author contributions

W.Ma and H.W. conceived and designed the study. W.Ma performed the analyses and wrote the initial draft of the paper. W.Ma, H.W., G.C., L.R.L., J.L., P.J.R., Q.F., B.K., Y.Z., J.J.C., and W.Maslowski contributed to interpreting the results, editing, and revising the manuscript.

## Competing interests

The authors declare no competing interests.
