## [Peer Review File · Nature Communications]

The Role of Interdecadal Climate Oscillations in Driving Arctic Atmospheric River TrendsREVIEWER COMMENTS

Reviewer #1 (Remarks to the Author):

Summary

In this study, Ma and coauthors analyze trends in Arctic atmospheric rivers (ARs) and assess the respective contributions of internal climate variability and anthropogenic warming to these trends. They find that AR frequency in the historical record has increased at a greater rate over the Atlantic sector relative to the Pacific sector of the Arctic. Using a suite of ensemble simulations from CESM2 and other CMIP6 models, they determine that internal climate variability in the form of decadal-scale phase shifts of the IPO and AMO have contributed to these disparate trends in Atlantic and Pacific sector ARs, with a more spatially uniform increase in AR frequency in free-running model simulations. They also use a decomposition method to clarify the role of dynamical and thermodynamic contributions to observed AR trends, finding that thermodynamic contributions are more important in most of the Arctic.

In my opinion, this paper is well-written and well-organized, and provides an important advance in scientific understanding of Arctic ARs and their large-scale coupled climate drivers. I have a number of minor comments detailed below, mainly requesting that the authors better situate their findings within the body of existing literature and assess the sensitivity of their results to the choice of AR detection algorithm. Provided these comments are addressed, I feel this paper will be an excellent contribution to the literature on Arctic ARs.

Minor comments

Have the authors confirmed that their results about AR trends are repeated across different AR algorithms? The ARTMIP project offers MERRA-2 AR datasets from a number of algorithms in addition to the Guan and Waliser algorithm used in this study. In particular, the requirement of meridional transport in the Guan and Waliser algorithm may affect the AR trend results near the North Pole (see comment below on L96–97).

- List of ARTMIP algorithms: <https://www.cgd.ucar.edu/projects/artmip/algorithms>

- ARTMIP "Tier 1" datasets:

<https://www.earthsystemgrid.org/dataset/ucar.cgd.artmip.tier1.catalogues.html>

Another way to check the robustness of the results would be to calculate trends in IVT magnitude, which does not depend on the AR algorithm chosen and does not flip sign across the North Pole. This would be a good complement to the TCWV trend shown in Fig. 1d and ED Fig. 3d.

Why are the maps cut off at 70N instead of the Arctic Circle (66.34N)?

L51–52: While I agree with the authors that a more systematic understanding of changes in ARs in the Arctic is needed, there are some recent studies that have examined trends in Arctic moisture transport and ARs. In particular, see Nygård et al. (2020) and Chen Zhang et al. (2023).

L64–66: Similarly, the increasing trend in ARs in the Atlantic sector has been well documented by Pengfei Zhang et al. (2023), and numerous studies have linked the enhanced sea ice decline in the Atlantic sector of the Arctic to increasing poleward moisture transport in this region. The authors should better situate their findings in the context of this previous literature. The finding of a greater AR increase in the Atlantic sector than the Pacific sector is certainly interesting but has more precedent in the literature than this manuscript seems to imply (e.g. in L64–66 and L281–282).

L96–97: What is the reason for the odd patterns across the North Pole in extended data Figure 2 (panels b–d)? Is this an artifact of the AR algorithm (in particular the meridional wind requirement)?

L121–125: It is nice to see that a large number of CMIP6 models were used beyond just the LENS2. This lends confidence to the results.

L220–229: Is there a role of the NAO in the dynamical contribution of the AMO to AR frequency? The out-of-phase pattern between a positive AMO influence in Baffin Bay and a negative influence in the Nordic seas (Fig. 5e) suggests there may be an interaction with the NAO. Previous studies (e.g. Liu and Barnes, 2015; Mattingly et al., 2018) have found that a negative NAO favors poleward moisture transport to the west of Greenland and a positive NAO favors transport to the east of Greenland. This pattern is also evident in the dynamical contributions to AR trends shown in extended data figures 2b (ERA5) and 3b (MERRA-2).

L262–267: Is the enhanced future role of the forced AR trend over the Pacific sector due to projected sea ice decline in this region? Similarly, is the near-future weakening of forced AR trends over the Greenland Sea due to the fact that sea ice has already declined significantly in this region, leaving less capacity for future sea ice decline?

- On a related note, it would strengthen the paper to discuss the specific spatial trends in Arctic sea ice loss and ocean warming in more detail. The authors state in L136–137 that "The historical SST/sea ice variability is key to understanding the observed pattern in Arctic AR trends", but do not describe the nature of these trends in the Arctic. See for example Årthun et al. 2012, Polyakov et al. 2020, Skagseth et al. 2020, and other papers on the "Atlantification" and "borealization" of the Arctic Ocean.

Technical corrections

L22: "multi sources" - use better phrase, e.g. "multiple sources"

- Also L63 ("multi-source data")

L36: "on"  "of"

L53: Choose a more appropriate word than "huge", e.g. "significant" or "major"

L60: "scale"  "scales"

L72–73: More precisely, a *sea ice*-free Arctic

L78: Change to "...a grid point *is* under AR *conditions*"

L104: Choose better word than "a little", e.g. "slightly"

L149–150: This sentence is grammatically incorrect - "variability" is a singular noun and doesn't agree with the plural "those" and "play important roles".

L154: "for *the* majority"

L179: Choose a better phrase than "it becomes immediately obvious" - perhaps "it is clear"

L210: Is this sentence referencing Fig. 5d rather than 5c? Please check figure references throughout the paper.

L211: "favors the increasing"  "favors increasing"

L212: "and the decreasing"  "and decreasing"

L250–251: "*the* SSP370 warming scenario"

L288–289: "phase... is"  "phases... are"

References

- Årthun, M., Eldevik, T., Smedsrud, L. H., Skagseth, Ø., & Ingvaldsen, R. B. (2012). Quantifying the Influence of Atlantic Heat on Barents Sea Ice Variability and Retreat. *Journal of Climate*, 25(13), 4736–4743. <https://doi.org/10.1175/JCLI-D-11-00466.1>
- Liu, C., & Barnes, E. A. (2015). Extreme moisture transport into the Arctic linked to Rossby wave breaking. *Journal of Geophysical Research: Atmospheres*, 120(9), 3774–3788. <https://doi.org/10.1002/2014JD022796>
- Nygård, T., Naakka, T., & Vihma, T. (2020). Horizontal moisture transport dominates the regional moistening patterns in the Arctic. *Journal of Climate*, 33(16), 6793–6807. <https://doi.org/10.1175/JCLI-D-19-0891.1>
- Polyakov, I. V., Alkire, M. B., Bluhm, B. A., Brown, K. A., Carmack, E. C., Chierici, M., et al. (2020). Borealization of the Arctic Ocean in Response to Anomalous Advection From Sub-Arctic Seas. *Frontiers in Marine Science*, 7, 491. <https://doi.org/10.3389/fmars.2020.00491>
- Skagseth, Ø., Eldevik, T., Årthun, M., Asbjørnsen, H., Lien, V. S., & Smedsrud, L. H. (2020). Reduced efficiency of the Barents Sea cooling machine. *Nature Climate Change*, 10(7), 661–666. <https://doi.org/10.1038/s41558-020-0772-6>
- Zhang, C., Tung, W., & Cleveland, W. S. (2023). Climatology and decadal changes of Arctic atmospheric rivers based on ERA5 and MERRA-2. *Environmental Research: Climate*, 2(3), 035005. <https://doi.org/10.1088/2752-5295/acdf0f>

Reviewer #2 (Remarks to the Author):

The manuscript presents an observational but mostly modeling study to explain the observed discrepancy between faster increasing atmospheric rivers (ARs) into the Arctic in the North Atlantic sector than the North Pacific sector despite that the models predict global warming will increase ARs into the Arctic more evenly between both ocean basins. The authors argue that a shift to the negative polarity of the Interdecadal Pacific Oscillation over the past forty years has dampened or canceled the frequency of ARs into the Arctic due to global warming. The authors then further argue that given the strong influence of the major natural modes of ocean variability on AR frequency into the Arctic, removing their influence from model projections forced by increasing greenhouse gases, will reduce the uncertainty surrounding the influence of global warming on AR frequency into the Arctic.

To my eye, in Extended Data Figure 5, the trend in global sea surface temperatures (SSTs) between 60°S to 70°N can be grossly described as global warming with a La Niña signal superimposed. Since the

positive phase of the Atlantic Multidecadal Oscillation (AMO) is mostly a warming of SSTs in the North Atlantic, the SST trends over the past forty years can be accurately described as a shift to the positive phase of the AMO. However, the trend in SSTs in the North Pacific, outside of the tropics is also broadly warming and hard to see how that projects strongly onto the negative phase of the Interdecadal Pacific Oscillation (IPO). A robust shift to the negative phase of the IPO might have occurred between 1990 and 2010 but not for the past forty years. I also think that the SSTs in the North Pacific shown in the second mode of the MCA in Figure 3d does not project onto the decadal trends in North Pacific SSTs. The authors stated that in the models (related to Figures 2 and 5) the frequency of ARs in the Arctic is highly sensitive to a warming atmosphere and that seems to me to be the simplest explanation of why in the second mode of the MCA, North Atlantic ARs into the Arctic are increasing due to higher atmospheric moisture in the ocean basin related to warmer SSTs while North Pacific ARs into the Arctic are decreasing due to lower atmospheric moisture in the ocean basin related to colder SSTs. I would further argue that this is consistent with Figure 4 and Extended Data Figure 6. The North Pacific trends in ARs into the Arctic are consistent with SSTs temperatures where warmer SSTs, especially at higher latitudes, favor an increase in ARs which resembles the SST trend shown in Extended Data Figure 5a and does not resemble the shift the negative phase of the IPO as shown in Figure 3d.

Based on the analysis presented, I feel that the authors have not demonstrated the physical reason for the observed discrepancy between the AR trends into the Arctic in the North Atlantic and North Pacific sectors. Based on the modeling results shown, it seems reasonable to attribute the increasing trend in ARs into the Arctic observed in the North Atlantic sector to warming North Atlantic SSTs and a shift to the positive phase of the AMO. In contrast I fail to see modeling evidence for the observed decreasing trend in ARs into the Arctic observed in the North Pacific sector.

In summary, I do think that this is an interesting study on an interesting and important topic. But I see some important flaws in the study. I think that the authors need to present more convincing support that the models demonstrate the negative phase of the IPO favors less ARs into the Arctic in the North Pacific sector (my one minor suggestion would be start), that the observed North Pacific SST trend strongly projects onto the negative phase of the IPO and why is there a lack of an increasing AR trend in the North Pacific despite warming SSTs.

I have one more minor comment below. The authors need to better support their arguments and conclusions and therefore I am recommending major revisions.

Minor comments:

1. Figure 2 – I think that the authors should show the trends for GOGA in addition to the difference in trends between LENS and GOGA.

Reviewer #3 (Remarks to the Author):

This paper shows that trends in basin-scale SST patterns affect trends in AR frequency over the Arctic. Ample and convincing evidence is shown to support this claim. This is useful knowledge, and so I think the paper makes useful contribution to the literature

on Arctic change. The paper is well structure and clearly written. I have only one substantial objection, but addressing it should only require some re-writing. I also have requests for some additional figures that would allow the reader to better appreciate the results; these do not requiring extensive new analysis.

1. There is one sense in which the paper is misleading, and this should be addressed before publication. Throughout the paper, an implicit assumption is made that ensemble-mean model trends faithfully capture the real-world response to anthropogenic forcing, and observed trends in IPO and AMO are pure unforced internal variability. There is no mention of the alternative hypothesis that much of the observed trends may in fact be a response to anthropogenic forcing, which models fail to capture. The paper provides no evidence or argument to exclude this alternative hypothesis. This issue should specifically be addressed at the following points in the paper:

I.67-68: "the discrepancy between the observed AR trends and the anthropogenically driven trends" -- since the observed trends could in fact be largely anthropogenically driven, it would be more accurate to rephrase to "the discrepancy between the observed AR trends and the model ensemble-mean trends"

I.126-127: "factors besides anthropogenic forcing likely play a role in shaping the spatially differing observed trend" -- this sentence is very generic, and wording implies that anthropogenic forcing cannot by itself produce the spatially differing trend, but this implication is not supported by any evidence provided here. Should reword to make the sentence more explicit: the observed trends could result purely from internal variability, a response to forcing not captured by models, or a combination of both. In any case, the real relevance of the paper is to show how specific SST patterns (whatever their ultimate cause) affect AR frequency.

I.149-150: "The above analyses suggest that internal variability, especially those pertaining to the ocean, play important roles in modulating Arctic AR trends at the interdecadal time scale". Again, the difference between observed and model-ensemble means is attributed purely to internal variability with no justification. Should more accurately reword to "The above analyses suggest that large-scale SST patterns play an important role in modulating Arctic AR trends at the interdecadal time scale"

I.285-286: "anthropogenic forcing alone leads to a uniform increase in ARs over the Arctic". Should be sharpened to "in climate models, anthropogenic forcing alone leads to a uniform increase in ARs over the Arctic"

I.287: "anthropogenically driven trends" should be replaced with "model ensemble-mean trends"

In view of the above, I also recommend changing the title to something that more accurately describes the substantiated content of the paper, namely "Understanding the

role of global SST patterns in driving ..." or similar.

2. l.131-133: It would be beneficial to also show the absolute GOGA trends in Fig 2 or in supplemental (not just difference from LENS) to appreciate how well GOGA captures the observed trend. It would also be good to show the full dynamical-thermodynamical decomposition of both LENS and GOGA to compare with Fig 1.

3. l.184-185: In Fig 4, I would suggest also showing the correlation between the IPO and Pacific-sector AR frequency, and between AMO and Atlantic-sector AR; this could be shown in same panels with different colors.

Minor comments, typos:

l. 24: the Arctic AR frequency -> the observed Arctic AR frequency

l. 43: "Recent studies have further revealed that 70 – 80% of the 44 atmospheric moisture transported into the Arctic is accomplished by ARs". This fact was earlier pointed out by Woods et al. 2013 and Liu and Barnes 2015, and suggest including these references.

Woods, C., R. Caballero, and G. Svensson, 2013: Large-scale circulation associated with moisture intrusions into the Arctic during winter. *Geophys. Res. Lett.*, 40, 4717–4721.

Liu, C., and E. A. Barnes, 2015: Extreme moisture transport into the Arctic linked to Rossby wave breaking. *J. Geophys. Res. Atmos.*, 120, 3774–3788.

l. 78: "the fraction of time a grid point under AR condition (in percentage)". Insert "is" before "under". Also please specify what is meant by "under AR conditions" (presumably that an AR is detected at a specific point in space-time).

l. 97-102: This material seems vague and speculative to me, and adds little substance to the paper; I suggest removing it altogether.

l.489: grids -> grid points. Also at l.520 and 522.

Reviewer: Rodrigo Caballero

NCOMMS-23-27686-T: Response to Review Comments

Reviewer #1

Summary

In this study, Ma and coauthors analyze trends in Arctic atmospheric rivers (ARs) and assess the respective contributions of internal climate variability and anthropogenic warming to these trends. They find that AR frequency in the historical record has increased at a greater rate over the Atlantic sector relative to the Pacific sector of the Arctic. Using a suite of ensemble simulations from CESM2 and other CMIP6 models, they determine that internal climate variability in the form of decadal-scale phase shifts of the IPO and AMO have contributed to these disparate trends in Atlantic and Pacific sector ARs, with a more spatially uniform increase in AR frequency in free-running model simulations. They also use a decomposition method to clarify the role of dynamical and thermodynamic contributions to observed AR trends, finding that thermodynamic contributions are more important in most of the Arctic.

In my opinion, this paper is well-written and well-organized, and provides an important advance in scientific understanding of Arctic ARs and their large-scale coupled climate drivers. I have a number of minor comments detailed below, mainly requesting that the authors better situate their findings within the body of existing literature and assess the sensitivity of their results to the choice of AR detection algorithm. Provided these comments are addressed, I feel this paper will be an excellent contribution to the literature on Arctic ARs.

We thank the reviewer for carefully reviewing our manuscript and providing the encouraging, positive comments. The constructive comments have greatly improved the quality of this manuscript. In particular, the suggestion on looking into the datasets from other AR algorithms that participated in ARTMIP improves the robustness of the results presented in this study. Our responses to the specific comments are shown below, with your original comments in blue.

Minor comments

Have the authors confirmed that their results about AR trends are repeated across different AR algorithms? The ARTMIP project offers MERRA-2 AR datasets from a number of algorithms in addition to the Guan and Waliser algorithm used in this study. In particular, the requirement of meridional transport in the Guan and Waliser algorithm may affect the AR trend results near the North Pole (see comment below on L96–97).

- List of ARTMIP algorithms: <https://www.cgd.ucar.edu/projects/artmip/algorithms>

- ARTMIP "Tier 1" datasets:

<https://www.earthsystemgrid.org/dataset/ucar.cgd.artmip.tier1.catalogues.html>

Another way to check the robustness of the results would be to calculate trends in IVT magnitude, which does not depend on the AR algorithm chosen and does not flip sign across the North Pole. This would be a good complement to the TCWV trend shown in Fig. 1d and ED Fig. 3d.

Thank you for these great suggestions. We have now calculated the trends in IVT magnitude. As shown in Figure R1, the trends of both mean IVT and extreme IVT magnitude show stronger increases over the Atlantic sector compared to the Pacific sector. In addition, we have also examined the Arctic AR frequency trends from 1980 to 2016 in the different AR datasets derived using various global AR detection algorithms that participated in ARTMIP. As shown in Figure R2 and R3, similar patterns of stronger AR increases over the Atlantic sector compared to the Pacific sector can also be identified in most of these algorithms, except for the ones that detect almost no AR over the Arctic. These results thus further corroborate the robustness of the findings presented in the manuscript. We have now properly included these results in the revised manuscript as follows.

“This stronger moistening over the Atlantic sector intensifies both the mean and extreme integrated water vapor transport (IVT) trends there and results in more frequent AR occurrence (Extended Data Fig. 1).”

“Furthermore, the spatial pattern in the Arctic AR frequency trend observed in this study is not sensitive to the use of different AR detection algorithms (Supplementary Fig. 3). This observed pattern can also be identified in the AR datasets derived using various global AR detection algorithms that participated in ARTMIP⁴⁸, except for the ones that detect almost no AR over the Arctic (Supplementary Fig. 4).”

Figure R1. Trends in (a) mean IVT and (b) extreme IVT magnitude based on ERA5 data from 1981 to 2021. Extreme IVT is defined as the 85th percentile of the daily IVT magnitude of each year.

Figure R2. Arctic AR frequency trends quantified with different AR detection algorithms that participate in ARTMIP. Trends are based on MERRA-2 data from 1980 to 2016.

Figure R3. Arctic AR frequency climatology quantified with different AR detection algorithms that participate in ARTMIP. Results are based on MERRA-2 3-hourly data from 1980 to 2016.

Why are the maps cut off at 70N instead of the Arctic Circle (66.34N)?

Although the Arctic is formally defined as regions poleward of 66.34°N, it is also very convenient and common to define the Arctic as regions poleward of 70°N in the literature of Arctic climate studies, especially for the comparison between coarse-grid climate models. We simply follow this convention in this study. Here are a few examples that used this definition:

Bintanja, R., Andry, O. Towards a rain-dominated Arctic. *Nature Clim Change* 7, 263–267 (2017). <https://doi.org/10.1038/nclimate3240>

Landrum, L., Holland, M.M. Extremes become routine in an emerging new Arctic. *Nature Clim Change* 10, 1108–1115 (2020). <https://doi.org/10.1038/s41558-020-0892-z>

McCrystall, M.R., Stroeve, J., Serreze, M. et al. New climate models reveal faster and larger increases in Arctic precipitation than previously projected. *Nat Commun* 12, 6765 (2021). <https://doi.org/10.1038/s41467-021-27031-y>

Screen, J., Simmonds, I. The central role of diminishing sea ice in recent Arctic temperature amplification. *Nature* 464, 1334–1337 (2010). <https://doi.org/10.1038/nature09051>

L51–52: While I agree with the authors that a more systematic understanding of changes in ARs in the Arctic is needed, there are some recent studies that have examined trends in Arctic moisture transport and ARs. In particular, see Nygård et al. (2020) and Chen Zhang et al. (2023).

Thank you for pointing us to these references. More discussions on the Arctic AR and moisture transport trends found in previous studies, including Nygård et al. (2020) and Zhang et al. (2023), have been added in the Introduction section.

“It has been shown that ARs or extreme moisture intrusions have been increasing over the Atlantic sector of the Arctic during winter^{32,37,41}. This increase in ARs contributes to the decline in sea ice over the Barents-Kara Sea³². The Arctic-wide annual AR counts have shown an upward trend in the past four decades, with the peak AR occurrence frequency shifting poleward from overland to the Arctic Ocean⁴².”

We have also revised the following sentence from “*However, a systematic understanding of how AR occurrence frequency over the Arctic has changed in recent decades is still lacking.*” to “*However, a more systematic understanding on the spatial distribution of trends in the Arctic AR occurrence frequency in recent decades and the associated driving mechanisms is still lacking.*”

L64–66: Similarly, the increasing trend in ARs in the Atlantic sector has been well documented by Pengfei Zhang et al. (2023), and numerous studies have linked the enhanced sea ice decline in the Atlantic sector of the Arctic to increasing poleward moisture transport in this region. The authors should better situate their findings in the context of this previous literature. The finding of a greater AR increase in the Atlantic sector than the Pacific sector is certainly interesting but has more precedent in the literature than this manuscript seems to imply (e.g. in L64–66 and L281–282).

We agree with the reviewer that previous studies have explored the increasing AR or moisture transport trends over the Atlantic sector. While those studies focus solely on the Atlantic sector, our study is the first one to contrast the different trends over the Pacific sector and the Atlantic sector and identify the mechanisms driving such differing spatial trends. Nonetheless, as mentioned above in the previous response, additional references have been included in the manuscript to acknowledge those studies that reported the observed increasing AR or moisture transport trends over the Atlantic sector of Arctic.

L96–97: What is the reason for the odd patterns across the North Pole in extended data Figure 2 (panels b–d)? Is this an artifact of the AR algorithm (in particular the meridional wind requirement)?

This is not an artifact of the AR algorithm. Extended Data Figure 2d in the original manuscript shows the summer 850 hPa meridional wind trend in ERA5. It is not the meridional wind trend associated with ARs. Thus, the pattern shown in Extended Data Figure 2d is not dependent on the AR detection algorithm used in this study. As shown in Figure R4 below, the negative meridional wind trend (more northerly) over the Atlantic sector is driven by the positive sea level pressure (SLP) trend over Greenland and the negative SLP trend over northern Eurasia. This negative meridional wind trend reduces ARs over the Atlantic sector (Extended Data Figure 2b in the original manuscript). On the other hand, the negative SLP trend over Northern Eurasia drives a positive meridional wind trend (more southerly) over the Laptev Sea and slightly enhances AR activities there (Extended Data Figure 2b in the original manuscript). The summer AR frequency trend due to dynamic contribution shown in Extended Data Figure 2b in the original manuscript is thus consistent with the meridional wind trends and the SLP trends. Extended Data Figure 2 in the original manuscript has been moved to the Supplementary Materials as Supplementary Figure 2. We have now included more clarification of this in the figure caption.

*“Note that the wind trends shown in **c** and **d** are the seasonal mean wind trends. Thus, their patterns are not dependent on the AR detection algorithm used.”*

Figure R4. Summer sea level pressure trend (shaded contours) and 850 mb horizontal wind vector trend in ERA5 from 1981 to 2021.

L121–125: It is nice to see that a large number of CMIP6 models were used beyond just the LENS2. This lends confidence to the results.

Thank you!

L220–229: Is there a role of the NAO in the dynamical contribution of the AMO to AR frequency? The out-of-phase pattern between a positive AMO influence in Baffin Bay and a negative influence in the Nordic seas (Fig. 5e) suggests there may be an interaction with the NAO. Previous studies (e.g. Liu and Barnes, 2015; Mattingly et al., 2018) have found that a negative NAO favors poleward moisture transport to the west of Greenland and a positive NAO favors transport to the east of Greenland. This pattern is also evident in the dynamical contributions to AR trends shown in extended data figures 2b (ERA5) and 3b (MERRA-2).

Great point! Indeed, the SLP pattern associated with the positive AMO shown in Extended Data Figure 7c in the original manuscript resembles the negative phase of Arctic Oscillation and also projects onto the negative phase of NAO over the North Atlantic. After a further literature review, we found previous studies showing that a positive AMO can drive the formation of negative NAO, especially during the cold season (e.g. Grossmann and Klotzbach, 2009; Gastineau and Frankignoul, 2012; Omrani et al., 2014). Positive AMO thus usually leads the

negative NAO by a few years. This suggests that NAO likely played a role in the dynamical contribution to AR frequency trends attributed to AMO phase change. However, since NAO is not independent of AMO in this case, it would be difficult to separately quantify the role NAO played in modulating the AR frequency trends. We have provided more discussions to point out a potential role of NAO in the dynamical contribution of the AMO to AR frequency.

“In response to the positive AMO, high SLP anomalies form over almost the entire Arctic, with negative SLP anomalies found over mid-latitude regions. This SLP anomaly pattern resembles the negative phase of Arctic Oscillation⁵⁰. The high SLP anomalies have two centers, including one located over the Laptev Sea and the other south of Iceland. The high SLP anomaly over the Laptev Sea extends eastward into the Beaufort Sea and induces northeasterly wind anomaly there (Extended Data Fig. 8c). A positive AMO thus acts to reduce ARs over the Beaufort Sea. Over the North Atlantic, the high SLP anomaly south of Iceland is accompanied by a low SLP anomaly further south. Consistent with previous studies^{51–53} which show that a positive AMO can induce a negative North Atlantic Oscillation (NAO), especially during the cold season, this dipole SLP anomaly pattern projects onto the negative phase of NAO⁵⁴. The negative NAO pattern enhances AR activities over the Baffin Bay and suppresses those over the Barents Sea (Fig. 5e), in line with the role of NAO in modulating poleward moisture transport⁵⁵.”

References:

Hurrell, J. W., Kushnir, Y., Ottersen, G. & Visbeck, M. An overview of the North Atlantic oscillation. *Geophys. Monogr. Geophys. Union* **134**, 1–36 (2003).

Grossmann, I., and Klotzbach, P. J. (2009), A review of North Atlantic modes of natural variability and their driving mechanisms, *J. Geophys. Res.*, 114, D24107, doi:[10.1029/2009JD012728](https://doi.org/10.1029/2009JD012728).

Gastineau, G., Frankignoul, C. Cold-season atmospheric response to the natural variability of the Atlantic meridional overturning circulation. *Clim Dyn* 39, 37–57 (2012). <https://doi.org/10.1007/s00382-011-1109-y>

Liu, C. & Barnes, E. A. Extreme moisture transport into the Arctic linked to Rossby wave breaking. *J. Geophys. Res. Atmos.* **120**, 3774–3788 (2015).

Mattingly, K. S., Mote, T. L. & Fettweis, X. Atmospheric river impacts on Greenland Ice Sheet surface mass balance. *J. Geophys. Res. Atmos.* **123**, 8538–8560 (2018).

Omrani, NE., Keenlyside, N.S., Bader, J. *et al.* Stratosphere key for wintertime atmospheric response to warm Atlantic decadal conditions. *Clim Dyn* 42, 649–663 (2014). <https://doi.org/10.1007/s00382-013-1860-3>

Thompson, D. W. J. & Wallace, J. M. The Arctic oscillation signature in the wintertime geopotential height and temperature fields. *Geophys. Res. Lett.* **25**, 1297–1300 (1998).

L262–267: Is the enhanced future role of the forced AR trend over the Pacific sector due to projected sea ice decline in this region? Similarly, is the near-future weakening of forced AR trends over the Greenland Sea due to the fact that sea ice has already declined significantly in this region, leaving less capacity for future sea ice decline?

- On a related note, it would strengthen the paper to discuss the specific spatial trends in Arctic sea ice loss and ocean warming in more detail. The authors state in L136–137 that "The historical SST/sea ice variability is key to understanding the observed pattern in Arctic AR trends", but do not describe the nature of these trends in the Arctic. See for example Årthun et al. 2012, Polyakov et al. 2020, Skagseth et al. 2020, and other papers on the "Atlantification" and "borealization" of the Arctic Ocean.

As shown in Figure R5a, an enhanced sea ice decline is found over large areas of the Pacific sector and along the sea ice edge regions over the Atlantic sector based on LENS2 (2024-2064). Indeed, large areas over the Atlantic sector experience a weaker sea ice decline in part because these regions have a lower sea ice concentration to start with. This spatial pattern in the sea ice concentration trends certainly contributes to the enhancement of forced AR trend over the Pacific sector and the weakening of forced AR trend over the Greenland Sea, as a stronger (weaker) sea ice decline can lead to more (less) moistening of the atmosphere associated with the local evaporation. However, differences exist between the spatial pattern of the sea ice concentration trends (Figure R5a) and the column-integrated water vapor trends (Figure R5b). For example, over the Atlantic sector, the weakest sea ice decline occurs over regions between 0°-30°E, while the weakest trend in the column-integrated water vapor is located over the Greenland Sea. Over the Pacific sector, the strongest column-integrated water vapor trend is found over south of the regions with the strongest sea ice decline. These differences between the spatial patterns of the sea ice trend and column-integrated water vapor trend suggest that sea ice decline is likely just one of the factors that contributes to the forced column-integrated water vapor trend and thus the forced AR trend during the future period (2024-2064). Since the investigation of future forced AR trends is not the focus of the current study, we will defer this science question to future research. Nevertheless, the spatial pattern of near-future forced AR trends shown in Figure 6a is consistent with that of the forced column-integrated water vapor trends shown in Figure R5b.

To provide more background on the observed Arctic sea ice and ocean warming trends, the following sentences have been added to the Introduction section:

“Concurrent with AA, the extent of Arctic sea ice has shown a substantial decline, with the strongest decline over the western Arctic during summer and over the Barents Sea during winter. While the summer western Arctic sea ice decline has been attributed to the recent persistent positive Pacific North American pattern, the strengthening and warming of the Atlantic inflow, which has been termed “Atlantification” of the Arctic Ocean, has warmed the Barents Sea and contributed to the winter sea ice decline there^{4,5}.”

Figure R5. Ensemble mean **a** sea ice concentration trends and **b** column-integrated water vapor trends in LENS2 from 2024 to 2064.

Technical corrections

L22: "multi sources" - use better phrase, e.g. "multiple sources"
 - Also L63 ("multi-source data")

We have revised them as suggested.

L36: "on"  "of"

Corrected.

L53: Choose a more appropriate word than "huge", e.g. "significant" or "major"

We have replaced "huge" with "significant".

L60: "scale"  "scales"

Corrected.

L72–73: More precisely, a *sea ice*-free Arctic

We have changed "ice-free Arctic" to "sea ice-free Arctic".

L78: Change to "...a grid point *is* under AR *conditions*"

We have changed the sentence to “*which is defined as the fraction of time (in percentage) when AR is detected at a grid point*” to better define AR frequency.

L104: Choose better word than "a little", e.g. "slightly"

We have replaced “a little” with “slightly”.

L149–150: This sentence is grammatically incorrect - "variability" is a singular noun and doesn't agree with the plural "those" and "play important roles".

This sentence has been changed to “*The above analyses suggest that large-scale SST patterns play an important role in modulating Arctic AR trends at the interdecadal time scale*”.

L154: "for *the* majority"

Corrected.

L179: Choose a better phrase than "it becomes immediately obvious" - perhaps "it is clear"

We have replaced “it becomes immediately obvious” with “it is clear”.

L210: Is this sentence referencing Fig. 5d rather than 5c? Please check figure references throughout the paper.

Yes, it should be Fig. 5d. Thank you for capturing this mistake. We have checked figure references throughout the paper.

L211: "favors the increasing"  "favors increasing"

Corrected.

L212: "and the decreasing"  "and decreasing"

Corrected.

L250–251: "*the* SSP370 warming scenario"

Corrected.

L288–289: "phase... is"  "phases... are"

Corrected.

References

Årthun, M., Eldevik, T., Smedsrud, L. H., Skagseth, Ø., & Ingvaldsen, R. B. (2012). Quantifying the Influence of Atlantic Heat on Barents Sea Ice Variability and Retreat. *Journal of Climate*, 25(13), 4736–4743. <https://doi.org/10.1175/JCLI-D-11-00466.1>

Liu, C., & Barnes, E. A. (2015). Extreme moisture transport into the Arctic linked to Rossby wave breaking. *Journal of Geophysical Research: Atmospheres*, 120(9), 3774–3788. <https://doi.org/10.1002/2014JD022796>

Nygård, T., Naakka, T., & Vihma, T. (2020). Horizontal moisture transport dominates the regional moistening patterns in the Arctic. *Journal of Climate*, 33(16), 6793–6807. <https://doi.org/10.1175/JCLI-D-19-0891.1>

Polyakov, I. V., Alkire, M. B., Bluhm, B. A., Brown, K. A., Carmack, E. C., Chierici, M., et al. (2020). Borealization of the Arctic Ocean in Response to Anomalous Advection From Sub-Arctic Seas. *Frontiers in Marine Science*, 7, 491. <https://doi.org/10.3389/fmars.2020.00491>

Skagseth, Ø., Eldevik, T., Årthun, M., Asbjørnsen, H., Lien, V. S., & Smedsrud, L. H. (2020). Reduced efficiency of the Barents Sea cooling machine. *Nature Climate Change*, 10(7), 661–666. <https://doi.org/10.1038/s41558-020-0772-6>

Zhang, C., Tung, W., & Cleveland, W. S. (2023). Climatology and decadal changes of Arctic atmospheric rivers based on ERA5 and MERRA-2. *Environmental Research: Climate*, 2(3), 035005. <https://doi.org/10.1088/2752-5295/acdf0f>

Reviewer #2

The manuscript presents an observational but mostly modeling study to explain the observed discrepancy between faster increasing atmospheric rivers (ARs) into the Arctic in the North Atlantic sector than the North Pacific sector despite that the models predict global warming will increase ARs into the Arctic more evenly between both ocean basins. The authors argue that a shift to the negative polarity of the Interdecadal Pacific Oscillation over the past forty years has dampened or canceled the frequency of ARs into the Arctic due to global warming. The authors then further argue that given the strong influence of the major natural modes of ocean variability on AR frequency into the Arctic, removing their influence from model projections forced by increasing greenhouse gases, will reduce the uncertainty surrounding the influence of global warming on AR frequency into the Arctic.

To my eye, in Extended Data Figure 5, the trend in global sea surface temperatures (SSTs) between 60°S to 70°N can be grossly described as global warming with a La Niña signal superimposed. Since the positive phase of the Atlantic Multidecadal Oscillation (AMO) is mostly a warming of SSTs in the North Atlantic, the SST trends over the past forty years can be accurately described as a shift to the positive phase of the AMO. However, the trend in SSTs in the North Pacific, outside of the tropics is also broadly warming and hard to see how that projects strongly onto the negative phase of the Interdecadal Pacific Oscillation (IPO). A robust shift to the negative phase of the IPO might have occurred between 1990 and 2010 but not for the past forty years. I also think that the SSTs in the North Pacific shown in the second mode of the MCA in Figure 3d does not project onto the decadal trends in North Pacific SSTs. The authors stated that in the models (related to Figures 2 and 5) the frequency of ARs in the Arctic is highly sensitive to a warming atmosphere and that seems to me to be the simplest explanation of why in the second mode of the MCA, North Atlantic ARs into the Arctic are increasing due to higher atmospheric moisture in the ocean basin related to warmer SSTs while North Pacific ARs

into the Arctic are decreasing due to lower atmospheric moisture in the ocean basin related to colder SSTs. I would further argue that this is consistent with Figure 4 and Extended Data Figure 6. The North Pacific trends in ARs into the Arctic are consistent with SSTs temperatures where warmer SSTs, especially at higher latitudes, favor an increase in ARs which resembles the SST trend shown in Extended Data Figure 5a and does not resemble the shift the negative phase of the IPO as shown in Figure 3d.

Based on the analysis presented, I feel that the authors have not demonstrated the physical reason for the observed discrepancy between the AR trends into the Arctic in the North Atlantic and North Pacific sectors. Based on the modeling results shown, it seems reasonable to attribute the increasing trend in ARs into the Arctic observed in the North Atlantic sector to warming North Atlantic SSTs and a shift to the positive phase of the AMO. In contrast I fail to see modeling evidence for the observed decreasing trend in ARs into the Arctic observed in the North Pacific sector.

In summary, I do think that this is an interesting study on an interesting and important topic. But I see some important flaws in the study. I think that the authors need to present more convincing support that the models demonstrate the negative phase of the IPO favors less ARs into the Arctic in the North Pacific sector (my one minor suggestion would be start), that the observed North Pacific SST trend strongly projects onto the negative phase of the IPO and why is there a lack of an increasing AR trend in the North Pacific despite warming SSTs.

I have one more minor comment below. The authors need to better support their arguments and conclusions and therefore I am recommending major revisions.

We thank the reviewer for providing constructive feedback on how to interpret the model results. Our point-by-point responses to each of the comments are shown below. For clarity, some of the original comments are quoted (in blue italic font).

We agree that the SST trend pattern shown in Extended Data Figure 5a (reproduced below in Figure R6a) can be approximately described as global warming with a La Niña signal superimposed. This is in part because: (1) we kept the global warming signal when calculating the SST trends and (2) the IPO has been transitioning to a positive phase since 2010 (Figure R6b). However, as described in the Method section, following Tokinaga et al. (2017), when calculating the IPO index, SST data must be first **detrended** to minimize the influences of the global warming signal. Detrending is usually a required step for calculating the IPO index when the effects of anthropogenic warming cannot be neglected (e.g., Tokinaga et al., 2017; Huang et al., 2020; Luo et al., 2022). In this study, the detrending is based on the observed SST data from 1870 to 2021. We then projected the **detrended** observed SST onto the IPO pattern to obtain the observed IPO index. As shown in Figure R6b, the observed IPO index has exhibited an **overall** (Line 163 in the original manuscript) significant negative trend since 1981 (P value = 0.003). Since the definition of IPO is always based on the **detrended** SST, there is no surprise that the observed SST trends, which contain the global warming signal, do not project strongly onto the negative phase of IPO. Therefore, for the same reasons as described above, we also do not expect that SST pattern in the North Pacific, shown in the second mode of the MCA based on the unforced SSTs (Figure 3d), projects onto the decadal trends in North Pacific SSTs shown in Figure R6a. An **overall** observed negative IPO trend in the past four decades has been well documented by numerous studies (e.g., Figure 1 in Vance et al., 2022; Figure 4 in Imada et al., 2016; Supplementary Figure 4 in Dong et al., 2021). To further demonstrate the robustness of the

overall observed negative IPO trend in the past four decades, we have plotted in Figure R7 the time series of IPO index downloaded directly from the NOAA Physical Sciences Laboratory, which is calculated based on a different IPO definition and a different SST dataset (see <https://psl.noaa.gov/data/timeseries/IPOTPI/> for more details). As shown in Figure R7, their observed IPO also shows a significant negative trend since 1981. Based on these results, we believe it is also reasonable to describe the observed SST trends, shown in Extended Data Figure 5a, as an **overall** negative IPO phase shift with a global warming signal superimposed.

Figure R6. a, Observed SST trends from 1981 to 2021 based on the HadSST data. Stippled areas indicate trends are significant at the 0.05 level based on the Student's t-test. **b**, Temporal evolution of the observed IPO index. **c**, Temporal evolution of the observed AMO index. The black lines show the linear trends of the IPO and AMO indices. Both trends are significant at the 0.05 level based on the Student's t-test.

Figure R7. Temporal evolution of the observed IPO index (red curve) and its linear trend (black line) from 1981 to 2016. The IPO index is obtained from the NOAA Physical Sciences Laboratory based on the SST dataset ERSST v5. Data after 02/2017 is not available at the time when the data was downloaded. The trend is significant at the 0.05 level based on the Student’s t-test. See <https://psl.noaa.gov/data/timeseries/IPOTPI/> for more details on how the IPO index is calculated.

The reviewer also commented that there is a lack of “*modeling evidence for the observed decreasing trend in ARs into the Arctic observed in the North Pacific sector*” and “*why is there a lack of an increasing AR trend in the North Pacific despite warming SSTs*”. We would like to clarify that the observed AR trends over the North Pacific sector have been increasing significantly over the past four decades, which is consistent with the warming SST over the North Pacific shown in Figure R6a. This can be seen clearly in Figure 1e of the manuscript. We also state this explicitly in the caption of Figure 1 and Lines 85-88 in the original manuscript as follows:

- “*The ensemble mean trends over both the Atlantic sector and Pacific sector are significant at the 0.05 level based on the Student’s t-test.*”
- “*Averaging over the Atlantic sector (red box in Fig. 1a) and the Pacific sector (magenta box in Fig. 1a) individually, ARs have been increasing at a rate of about 0.42 (0.49) % decade⁻¹ and 0.19 (0.29) % decade⁻¹ over the respective regions in ERA5 (MERRA-2) (Fig. 1e).*”

Regarding the key finding of spatial difference of AR trends in the manuscript, we are trying to demonstrate that the observed negative IPO trend and the positive AMO trend have worked together to offset the increasing AR trends over the Pacific sector due to anthropogenic forcing. Our results suggest that the anthropogenic forcing had a larger influence on the AR trends than AMO and IPO, leading to a significant positive net AR trend over the Pacific sector, although the

observed IPO and AMO variability did dampen the externally forced AR trends over the Pacific sector and make them weaker than the trends over the Atlantic sector.

Another comment the reviewer made is that more support is needed to show “*that the models demonstrate the negative phase of the IPO favors less ARs into the Arctic in the North Pacific sector*”. In the original manuscript, at least two lines of evidence have been provided to show how the negative IPO favors less ARs into the Arctic in the Pacific sector. We first demonstrate this point in Figure 3 through MCA. As shown in Figure 3d, a negative IPO-like SST trend pattern covaries strongly with negative AR trends over the Pacific sector of the Arctic (Figure 3c), suggesting that the negative phase of the IPO likely favors less ARs moving into the Arctic through the North Pacific sector. The reviewer has also nicely provided an explanation for the negative AR trends over the Pacific sector associated with the negative IPO-like SST trend pattern by saying that “*the simplest explanation of why in the second mode of the MCA, North Atlantic ARs into the Arctic are increasing due to higher atmospheric moisture in the ocean basin related to warmer SSTs while North Pacific ARs into the Arctic are decreasing due to lower atmospheric moisture in the ocean basin related to colder SSTs*”. To further consolidate the relationship between the negative AR trends over the Pacific sector and the negative IPO phase shift and validate the reviewer’s explanation, we like to point to the mechanisms of the IPO in driving AR changes over the Arctic illustrated in Figure 5 and Extended Data Figure 7 in the original manuscript. As shown in Figure 5a, the positive IPO favors more ARs over the Pacific sector. This is achieved by moistening of the atmosphere over the Pacific sector of the Arctic through the warmer SST or surface temperatures, especially at higher latitudes (Extended Data Figure 7b in the original manuscript and Figure R8 below). The moistened atmosphere ultimately leads to an increase in ARs over the Pacific sector (Figure 5c). These results are also consistent with the reviewer’s explanation: “*I would further argue that this is consistent with Figure 4 and Extended Data Figure 6. The North Pacific trends in ARs into the Arctic are consistent with SSTs temperatures where warmer SSTs, especially at higher latitudes, favor an increase in ARs...*” In our mind, if the reviewer agrees that warmer SSTs over higher latitudes during the positive phase of IPO shown in Figure 4 and Extended Data Figure 6 in the original manuscript favor an increase in ARs, in corollary cooler SSTs over higher latitudes during the negative phase of IPO favor less ARs into the Arctic.

To more explicitly demonstrate how a negative IPO modulates ARs and moisture over the Arctic, we processed additional AR data based on the ACCESS and CNRM model ensembles and plotted their AR and integrated water vapor (IWV) patterns associated with a negative phase of IPO (obtained by regressing the AR and IWV time series onto the negative IPO index). Figure R9 shows that the negative IPO can lead to reductions of ARs over the Pacific sector (Figure R9a, b, and c) due to the drying of the atmosphere (Figure R9d, e, and f) in all three large ensembles. At the same time, the negative IPO favors the increases of ARs over the Atlantic sector in both the LENS2 and ACCESS ensembles. As we have explained in the manuscript, the increases in ARs over the Atlantic sector are driven by the changes in circulation while the contribution from the moisture changes is negligible. In contrast, the negative IPO in the CNRM ensemble drives a reduction in ARs over the Atlantic sector between 30°W and 30°E, while the increases in ARs confine only over a small region between 30°E and 60°E (Figure R9c). The reduction in ARs over the Atlantic sector in CNRM is caused by the atmospheric drying there with a magnitude comparable to the drying over the Pacific sector (Figure R9d). As we will

show below, the CNRM ensemble likely over-represents the drying effect over the Atlantic sector associated with the negative IPO and thus exaggerates its effect on the AR reduction there. We further examine IWV patterns associated with the negative IPO in the CMIP6 ensemble and three additional single model ensembles. As clearly shown in Figure R10, the negative IPO can cause a significant drying over the Pacific sector while its effect over the Atlantic sector is negligible or even slightly positive (Figure R10b and d). The results presented in Figure R9 and R10 based on six single model ensembles and one multi-model ensemble clearly and robustly show that a negative IPO can drive an atmospheric drying over the Pacific sector and lead to a reduction in ARs there.

Although this is the first study on how the IPO modulates ARs over the Arctic, the effect of negative (positive) IPO on high-latitude North Pacific cooling (warming) has been well documented in the literature (e.g., Supplementary Figure 2 in Dong et al., 2021; Figure 2 in Villamayor and Mohino 2015; Figure 2 in Hu et al., 2018). In the manuscript, we demonstrate how the positive IPO favors more ARs into the Arctic in the North Pacific sector. We believe that flipping the sign in Figure 5a-c and Extended Data Figure 7a-b in the original manuscript would show how a negative IPO favors less ARs into the Arctic in the North Pacific sector through a relative cooling of the high-latitude North Pacific. Based on these results, we believe that we have shown substantial and compelling evidence, which is also consistent with the reviewer’s reasoning and argument, to support the statement that a negative phase of the IPO favors less ARs into the Arctic in the North Pacific sector. This perspective is also in alignment with the assessment made by Reviewer #3.

Figure R8. Surface temperature regression patterns associated with IPO in LENS2. Stippled areas indicate the regression anomalies are significant at the 0.05 level based on the Student’s t-test.

Figure R9. Ensemble mean AR frequency regression patterns associated with the negative phase of IPO in (a) LENS2, (b) ACCESS, and (c) CNRM. d-f, same as a-c, but for the ensemble mean IWV regression patterns. Regressions for LENS2 and ACCESS are based on data from 1979 to 2100, but from 1850 to 2014 for CNRM. Note that only 26 members are employed for CNRM because data prior to 1950 are not available in four of the members (member # 7, 8, 9, and 10). Stippled areas indicate the regression anomalies are significant at the 0.05 level based on the Student's t-test.

Figure R10. Ensemble mean IWV regression patterns associated with the negative phase of IPO in (a) CMIP6 multi-model ensemble, (b) MPI-ESM1-2-LR, (c) IPSL-CM6A-LR, and (d) HadGEM3-GC31-LL. The regression for the CMIP6 ensemble is based on the piControl experiment data from 19 out of the 23 models used in the CMIP6 ensemble in the original manuscript. We exclude IITM-ESM and TaiESM1 because these two models do not have all the required variables for the regressions (IITM-ESM does not have monthly IWV while TaiESM1 does not have monthly SST). We also exclude EC-Earth3 and INM-CM4-8. The piControl run in EC-Earth3 shows a very strong cooling over the Northern Hemisphere high latitude oceans, especially over the North Atlantic (Figure R11b). As a result, its SST pattern associated with the negative IPO shows very strong cooling over the Northern Hemisphere high latitude oceans, especially over the North Atlantic (Figure R11a). For INM-CM4-8, its piControl climate drifts toward a negative IPO-like SST pattern (Figure R11d). Consequently, its first EOF mode simply reflects the

overall trends in the SST (Figure R11c). Most of the models in the CMIP6 ensemble provide at least 500 years of data for their piControl runs. For the models with at least 500 years of data, the first 500 years are used. For a few of the models with less than 500 years of data, we simply use all the available data. The models with less than 500 years of data include: BCC-ESM1 (451 years), CESM2-WACCM (499 years), and IPSL-CM5A2-INCA (250 years). Only members with the variant index “r1i1p1f1” are used in the CMIP6 ensemble. The regressions for the 30-member MPI-ESM1-2-LR ensemble, 33-member IPSL-CM6A-LR ensemble, and the 40-member HadGEM3-GC31-LL ensemble are based on data from 1979 to 2100, from 1850 to 2014, and from 1850 to 2014, respectively. All regressions are based on monthly data. Stippled areas indicate the regression anomalies are significant at the 0.05 level based on the Student’s t-test.

Figure R11. Sea surface temperature (SST) anomalies associated with the first EOF SST mode over the Pacific in (a) EC-Earth3 and (c) INM-CM4-8. **b** and **d** show the SST trend in the piControl experiments of EC-Earth3 and INM-CM4-8, respectively. Stippled areas indicate the regression anomalies or trends are significant at the 0.05 level based on the Student’s t-test.

Minor comments:

1. Figure 2 – I think that the authors should show the trends for GOGA in addition to the difference in trends between LENS and GOGA.

Thank you for the suggestion. We have updated Figure 2 to include the historical AR trends in GOGA (Figure R12c).

Figure R12. Same as Figure 2 in the manuscript, but with the ensemble mean historical AR trends in GOGA plotted in c.

References:

Tokenaga, H., Xie, S.-P. & Mukougawa, H. Early 20th-century Arctic warming intensified by Pacific and Atlantic multidecadal variability. *Proc. Natl. Acad. Sci.* **114**, 6227–6232 (2017).

Xin Huang et al., South Asian summer monsoon projections constrained by the interdecadal Pacific oscillation. *Sci. Adv.* 6, eaay6546(2020). DOI:10.1126/sciadv.aay6546

Luo, B., Luo, D., Dai, A. *et al.* The modulation of Interdecadal Pacific Oscillation and Atlantic Multidecadal Oscillation on winter Eurasian cold anomaly via the Ural blocking change. *Clim Dyn* 59, 127–150 (2022). <https://doi.org/10.1007/s00382-021-06119-7>

Vance, T.R., Kiem, A.S., Jong, L.M. *et al.* Pacific decadal variability over the last 2000 years and implications for climatic risk. *Commun Earth Environ* 3, 33 (2022). <https://doi.org/10.1038/s43247-022-00359-z>

Imada, Y., Tatebe, H., Watanabe, M. *et al.* South Pacific influence on the termination of El Niño in 2014. *Sci Rep* 6, 30341 (2016). <https://doi.org/10.1038/srep30341>

Dong, L., Leung, L.R., Song, F. *et al.* Uncertainty in El Niño-like warming and California precipitation changes linked by the Interdecadal Pacific Oscillation. *Nat Commun* 12, 6484 (2021). <https://doi.org/10.1038/s41467-021-26797-5>

Villamayor, J. and Mohino, E. (2015), Robust Sahel drought due to the Interdecadal Pacific Oscillation in CMIP5 simulations. *Geophys. Res. Lett.*, 42: 1214–1222. doi: [10.1002/2014GL062473](https://doi.org/10.1002/2014GL062473)

Hu, Z., A. Hu, and Y. Hu, 2018: Contributions of Interdecadal Pacific Oscillation and Atlantic Multidecadal Oscillation to Global Ocean Heat Content Distribution. *J. Climate*, **31**, 1227–1244, <https://doi.org/10.1175/JCLI-D-17-0204.1>.

Reviewer #3

This paper shows that trends in basin-scale SST patterns affect trends in AR frequency over the Arctic. Ample and convincing evidence is shown to support this claim. This is useful knowledge, and so I think the paper makes useful contribution to the literature on Arctic change. The paper is well structure and clearly written. I have only one substantial objection, but addressing it should only require some re-writing. I also have requests for some additional figures that would allow the reader to better appreciate the results; these do not requiring extensive new analysis.

Thank you for the insightful and constructive feedback and suggestions, which are very valuable in helping us improve the quality and clarity of our manuscript. Our responses to each of your comments are shown below, with your original comments in blue.

1. There is one sense in which the paper is misleading, and this should be addressed before publication. Throughout the paper, an implicit assumption is made that ensemble-mean model trends faithfully capture the real-world response to anthropogenic forcing, and observed trends in IPO and AMO are pure unforced internal variability. There is no mention of the alternative hypothesis that much of the observed trends may in fact be a response to anthropogenic forcing, which models fail to capture. The paper provides no evidence or argument to exclude this alternative hypothesis. This issue should specifically be addressed at the following points in the paper:

l.67-68: "the discrepancy between the observed AR trends and the anthropogenically driven trends" -- since the observed trends could in fact be largely anthropogenically driven, it would be more accurate to rephrase to "the discrepancy between the observed AR trends and the model ensemble-mean trends"

We agree that we have been making an implicit assumption in this study that the models can faithfully capture the real-world climate response to the anthropogenic forcing, based on the credibility of CMIP6 models, and thus it is reasonable to treat the model ensemble mean response as the true response to anthropogenic forcing. However, we also think that the alternative hypothesis pointed out by the reviewer is plausible. We have made the suggested changes to account for this alternative hypothesis.

1.126-127: "factors besides anthropogenic forcing likely play a role in shaping the spatially differing observed trend" -- this sentence is very generic, and wording implies that anthropogenic forcing cannot by itself produce the spatially differing trend, but this implication is not supported by any evidence provided here. Should reword to make the sentence more explicit: the observed trends could result purely from internal variability, a response to forcing not captured by models, or a combination of both. In any case, the real relevance of the paper is to show how specific SST patterns (whatever their ultimate cause) affect AR frequency.

We have made this sentence more explicit by changing it to “*Such discrepancies between the observed and the simulated trends suggest that factors, such as the observed internal variability and/or model deficiency in capturing the forced response, likely play a role in shaping the spatially differing observed trend*”.

1.149-150: "The above analyses suggest that internal variability, especially those pertaining to the ocean, play important roles in modulating Arctic AR trends at the interdecadal time scale". Again, the difference between observed and model-ensemble means is attributed purely to internal variability with no justification. Should more accurately reword to "The above analyses suggest that large-scale SST patterns play an important role in modulating Arctic AR trends at the interdecadal time scale"

We have made the suggested change.

1.285-286: "anthropogenic forcing alone leads to a uniform increase in ARs over the Arctic". Should be sharpened to "in climate models, anthropogenic forcing alone leads to a uniform increase in ARs over the Arctic"

We have made the suggested change.

1.287: "anthropogenically driven trends" should be replaced with "model ensemble-mean trends"

We have made the suggested change.

In view of the above, I also recommend changing the title to something that more accurately describes the substantiated content of the paper, namely "Understanding the role of global SST patterns in driving ..." or similar.

We have changed the title to “Understanding Roles of Interdecadal Climate Oscillations in Driving the Observed Arctic Atmospheric River Trends”.

2. 1.131-133: It would be beneficial to also show the absolute GOGA trends in Fig 2 or in supplemental (not just difference from LENS) to appreciate how well GOGA captures the observed trend. It would also be good to show the full dynamical-thermodynamical

decomposition of both LENS and GOGA to compare with Fig 1.

Thank you for these suggestions. We have updated Figure 2 to include the ensemble mean historical AR trends in GOGA (Figure R13c). The ensemble mean AR trends in LENS2 due to dynamical changes and the ensemble mean AR trends in GOGA due to dynamical changes (Figure R14b) and thermodynamical changes (Figure R14c) are also plotted below and included as Extended Data Figure 4 in the revised manuscript. Additional discussion on the AR trends due to dynamical changes is also provided in the revised manuscript.

“GOGA also simulates a negative AR trend over the Atlantic sector due to dynamical changes (Extended Data Fig. 4b), which is consistent with observations (Fig. 1b).”

Figure R13. Same as Figure 2 in the manuscript, but with the ensemble mean historical AR trends in GOGA simulations plotted in c.

Figure R14. **a**, Ensemble mean Arctic AR frequency trend in LENS2 due to dynamical changes. **b**, Ensemble mean Arctic AR frequency trend in GOGA due to dynamical changes. **c**, Ensemble mean Arctic AR frequency trend in GOGA due to thermodynamical changes. The stippled areas indicate trends that are significant at the 0.05 level based on the Student's t-test.

3. 1.184-185: In Fig 4, I would suggest also showing the correlation between the IPO and Pacific-sector AR frequency, and between AMO and Atlantic-sector AR; this could be shown in same panels with different colors.

Thank you for the suggestion. We have created such a figure (Figure R15) and included it as Extended Data Figure 6 in the revised manuscript. Compared to the correlation between Arctic mean AR trends and the IPO trends and the correlation between the Arctic mean AR trends and the AMO trends, the positive correlation between Pacific sector mean AR trends and the IPO trends weakens, but the correlation between the Atlantic sector mean AR trends and the AMO trends strengthens. The weaker correlation shown in Figure R15a is partly caused by an outlier member that shows a slightly negative AR trend over the Pacific sector. Removing this member can increase the correlation to about 0.3, which is significant at the 0.05 level. Another possible explanation for this weaker correlation is that, as the regional focus moves from the entire Arctic

to the Pacific sector with a smaller area extent, other internal variability processes, such as atmospheric internal variability (e.g., Liu et al., 2021; Ding et al., 2019), likely play increasingly more important roles in modulating the interdecadal variability of ARs over this specific region. This can lead to a weaker correlation between the AR trends and the IPO trends. Indeed, if we make the region defined as the Pacific sector slightly larger from 70°N-80°N and 150°E-210°E to 70°N-80°N and 150°E-250°E, the correlation between IPO trends and the Pacific sector mean AR trends increases to about 0.29, which is also significant at the 0.05 level. For the Atlantic sector, it is directly connected to the North Atlantic. The influences from the North Atlantic can thus be felt more directly and strongly over the Arctic Atlantic sector. Further research is needed to understand the relative roles of the IPO and AMO versus other internal variability in controlling the AR variability within their respective Arctic sector. The following discussions have been added to the revised manuscript.

“Similar relationships also can be found between the mean AR trends in the Pacific sector (magenta box in Fig. 1a) and the IPO trends, as well as between the mean AR trends in the Atlantic sector (red box in Fig. 1a) and the AMO trends (Extended Data Fig. 6). However, the correlation for the former weakens and for the latter strengthens. This weakened correlation between the mean AR trends in the Pacific sector and the IPO trends is partly caused by an outlier member with a slightly negative AR trend over the Pacific sector. Removing this member can increase the correlation to about 0.3, which is significant at the 0.05 level. It may also be attributed to the relatively smaller area extent of the Pacific sector than the entire Arctic. When transitioning the regional focus from the entire Arctic to the Pacific sector alone, other internal variability processes, such as atmospheric internal variability^{33,49}, likely play increasingly more important roles in modulating the interdecadal variability of ARs over this specific region. The relative roles of the IPO versus other atmospheric internal variability in modulating the AR variability over the Pacific sector at different time scales warrant further studies.”

Figure R15. a, Scatterplots between the Pacific sector mean AR frequency trends and the IPO trends, where the red lines show the regression of data points for 50 members of LENS2. **b** same as in **a**, but for the Atlantic sector AR trends and AMO trends.

Minor comments, typos:

l. 24: the Arctic AR frequency -> the observed Arctic AR frequency

We have made the suggested change.

l. 43: "Recent studies have further revealed that 70 – 80% of the 44 atmospheric moisture transported into the Arctic is accomplished by ARs". This fact was earlier pointed out by Woods et al. 2013 and Liu and Barnes 2015, and suggest including these references.

Woods, C., R. Caballero, and G. Svensson, 2013: Large-scale circulation associated with moisture intrusions into the Arctic during winter. *Geophys. Res. Lett.*, 40, 4717–4721.

Liu, C., and E. A. Barnes, 2015: Extreme moisture transport into the Arctic linked to Rossby wave breaking. *J. Geophys. Res. Atmos.*, 120, 3774–3788.

These two references have been properly cited.

l. 78: "the fraction of time a grid point under AR condition (in percentage)". Insert "is" before "under". Also please specify what is meant by "under AR conditions" (presumably that an AR is detected at a specific point in space-time).

We have changed the sentence to “*which is defined as the fraction of time (in percentage) when AR is detected at a grid point*” to better define AR frequency.

l. 97-102: This material seems vague and speculative to me, and adds little substance to the paper; I suggest removing it altogether.

These sentences have been removed to make the paragraph more concise. The Extended Data Fig. 2 in the original manuscript has been moved to the Supplementary.

l.489: grids -> grid points. Also at l.520 and 522.

Corrected.

References:

Liu, Z., Risi, C., Codron, F. *et al.* Acceleration of western Arctic sea ice loss linked to the Pacific North American pattern. *Nat Commun* 12, 1519 (2021). <https://doi.org/10.1038/s41467-021-21830-z>

Ding, Q., Schweiger, A., L’Heureux, M. *et al.* Fingerprints of internal drivers of Arctic sea ice loss in observations and model simulations. *Nature Geosci* 12, 28–33 (2019). <https://doi.org/10.1038/s41561-018-0256-8>

REVIEWERS' COMMENTS

Reviewer #1 (Remarks to the Author):

Thanks to the authors for their thorough responses to mine and the other reviewers' comments. It is nice to see that the ARTMIP algorithms generally agree on the spatial pattern of Arctic AR trends in the historical record and with the spatial patterns in IVT, which lends further confidence to the study results.

I have one remaining minor comment as a follow-up to the discussion of my original comment on L96–97: I suggest the authors change the color ramp in Supplementary Figs. 2c & 2d to something other than a green/brown diverging color scheme (perhaps a red/yellow or red/white diverging scheme such as the one used on other figures in this study). This would help distinguish the wind-related variables from the moisture-related variables in the figure.

Besides this minor comment, I feel this paper is ready for publication and will provide an excellent contribution to the literature.

Reviewer #2 (Remarks to the Author):

The revised manuscript presents an observational but mostly modeling study to explain the observed discrepancy between faster increasing atmospheric rivers (ARs) into the Arctic in the North Atlantic sector than the North Pacific sector despite that the models predict global warming will increase ARs into the Arctic more evenly between both ocean basins. The authors argue that a shift to the negative polarity of the Interdecadal Pacific Oscillation over the past forty years has dampened or canceled the frequency of ARs into the Arctic due to global warming. The authors then further argue that given the strong influence of the major natural modes of ocean variability on AR frequency into the Arctic, removing their influence from model projections forced by increasing greenhouse gases, will reduce the uncertainty surrounding the influence of global warming on AR frequency into the Arctic.

I don't understand why there is both supplemental figures and extended data?

Otherwise, I thought that the authors did a good job responding to the reviewers and I am satisfied with the revised manuscript and therefore recommending acceptance.

Reviewer #3 (Remarks to the Author):

The authors have responded thoroughly and appropriately to all comments from the first round of review, and I have no further suggestions. I think the manuscript is ready for publication at this point.

NCOMMS-23-27686A: Response to Review Comments

We thank all reviewers for the great efforts they invested in reviewing our manuscript. The constructive comments provided by all reviewers have greatly improved the quality of this work. We have made additional revisions to the manuscript to address the remaining comments made by the reviewers.

In response to reviewer #1's comment on the color scheme used for Supplementary Figs. 2c & 2d in the original manuscript (Supplementary Figs. 3c & 3d in the revised manuscript), we have replaced the original green/brown diverging color scheme with red/blue diverging color scheme (Fig. R1).

Figure R1. Seasonal Arctic atmospheric river (AR) frequency trends due to dynamic contribution in ERA5. a-b, winter (December-February) and summer (June-August) Arctic AR frequency trends due to dynamic contribution. **c,** 850 mb zonal wind trend in winter. **d,** 850 mb meridional wind trend in summer. Note that the wind trends shown in **c** and **d** are the seasonal mean wind trends. Thus, their patterns are not dependent on the AR detection algorithm used. Stippled areas indicate trends are significant at the 0.05 level based on the Student's t-test.

In response to reviewer #2's comment on why our manuscript contains both supplementary figures and extended data figures, we have now moved all extended data figures to the Supplementary Information.